# Self-organizing actin patterns shape membrane architecture but not cell mechanics

M. Fritzsche[1], D. Li[2], H. Colin-York[1], V.T. Chang[1,3], E. Moeendarbary[4,5], J.H. Felce[1], E. Sezgin[1], G. Charras[6], E. Betzig[7] & C. Eggeling[1]

Cell-free studies have demonstrated how collective action of actin-associated proteins can organize actin filaments into dynamic patterns, such as vortices, asters and stars. Using complementary microscopic techniques, we here show evidence of such self-organization of the actin cortex in living HeLa cells. During cell adhesion, an active multistage process naturally leads to pattern transitions from actin vortices over stars into asters. This process is primarily driven by Arp2/3 complex nucleation, but not by myosin motors, which is in contrast to what has been theoretically predicted and observed *in vitro*. Concomitant measurements of mechanics and plasma membrane fluidity demonstrate that changes in actin patterning alter membrane architecture but occur functionally independent of macroscopic cortex elasticity. Consequently, tuning the activity of the Arp2/3 complex to alter filament assembly may thus be a mechanism allowing cells to adjust their membrane architecture without affecting their macroscopic mechanical properties.

[1] MRC Human Immunology Unit, Weatherall Institute of Molecular Medicine, University of Oxford, Headley Way, Oxford OX3 9DS, UK. [2] National Laboratory of Biomacromolecules, Institute of Biophysics, Chinese Academy of Sciences, Beijing 100101, China. [3] Division of Structural Biology, Wellcome Trust Centre for Human Genetics, University of Oxford, Oxford OX3 7BN, UK. [4] Department of Biological Engineering, Massachusetts Institute of Technology, Cambridge, Massachusetts 02142, USA. [5] Department of Mechanical Engineering, University College London, London WC1E 7JE, UK. [6] London Centre for Nanotechnology and Department of Cell & Developmental Biology, University College London, 17-19 Gordon Street, London WC1H 0AH, UK. [7] Howard Hughes Medical Institute, Janelia Research Campus, 19700 Helix Drive, Ashburn, Virginia 20147, USA. Correspondence and requests for materials should be addressed to M.F. (email: marco.fritzsche@rdm.ox.ac.uk) or to C.E. (email: christian.eggeling@rdm.ox.ac.uk).

Vital cellular processes such as cell migration, division and homeostasis rely on the active reorganization of the cortical actin network located underneath the plasma membrane[1,2]. A thorough understanding of the dynamic potential of this actin cortex notably requires knowledge of the underlying organization principles. Two fundamentally different mechanisms exist to generate macromolecular structures in cells such as the actin cortex: self-assembly and self-organization. Self-assembly involves the physical association of molecules into an equilibrium structure[3]. Self-organization requires the physical interaction of molecules far from equilibrium driven by the constant input of energy in a steady-state structure[4,5]. Consequently, the cellular actin cortex is compatible with the concept of self-organization.

The actin cortex is a complex system that comprises polydisperse actin filaments undergoing continuous turnover with constant growth of the filaments at their barbed ends and shrinkage at their pointed ends[6]. On one hand, the filaments are crosslinked over finite periods of time and redistributed by the action of molecular motors, such as myosin-II[7,8]. On the other hand, two filament subpopulations compose the cortex in eukaryotic cells[9,10]. In HeLa cells, they have 20-fold differing turnover rates of the actin monomers and arise from distinct nucleation pathways[9]: (1) polymerization of long filamentous actin (F-actin) driven by formin proteins, which associate with the fast-growing barbed end of actin; and (2) branching of short F-actin driven by the Arp2/3 complex, which binds to preexisting F-actin and nucleates new filaments. The latter population has been shown to account for 80% of the total F-actin in different cell types, such as cervical HeLa, Melanoma and T cells[6,9,10].

Filament crosslinking by myosin-II as well as the fact that the Arp2/3 complex nucleates new filaments at a distinct angle of 70° from its mother filament has direct implications on the actin cytoskeletal ultra-structure (such as the cortex mesh-size), the mechanical stress and consequently the dynamics of the actin cortex[6,11]. Theoretical considerations predict that changes in the mechanical stress of an active actin network can result in self-organization[12,13], specifically in the emergence of ordered architectures, such as vortices, asters and stars (Fig. 1). Vortices are ring-like structures with high rotational symmetry. Actin stars and asters are both characterized by an asterisk-like topology with distinct nucleation centres and arm-like radially oriented F-actin strands. A common notion is that stars are less frequent and significantly larger than asters and that the arms radiating from the centre of actin stars consist of multiple parallel bundled F-actin rather than individual actin filaments in asters[14–16].

Formation of these patterns is achieved by polarity sorting of actin filaments. Such sorting follows from the polarized turnover dynamics of F-actin and from the unidirectional motion of the myosin motors along the filaments they cross-link[12,16] and/or the nucleation activity of the Arp2/3 complex[14]. These processes become dominant when the intrinsic mechanical stress within the cortical network (for example, through the crosslinker or nucleator activity) overcomes a threshold value such as a critical force per unit volume, so that the reorganization of actin in the form of vortices, asters or stars are energetically favoured. In this process, randomly structured F-actin spontaneously rearranges (for example, rotates) into these structures[13–16]. Specifically, kinetic Monte Carlo simulations found that actin bundles are the starting point for the formation of actin stars. For example, asters and stars are predicted to form when the bending energy associated with bringing two filaments into contact is compensated by the energetic gain resulting from their crosslinking energy[15]. It is further predicted that, out of these bundles, nucleation activity of the Arp2/3 complex results in the formation of asters, which—in the presence of the actin-filament

crosslinker protein fascin—transform into stars, depending on the ratio of fascin and actin monomers (G-actin)[15]. This is supported by active gel theory, which from symmetry considerations predicts that, through continuous consumption of ATP, polar filaments induce a non-equilibrium state. This state is characterized by the generation of flows of molecules such as G-actin monomers or fascin and by mechanical stresses on the filaments[7,13].

In summary, in silico studies predict that actin bundles in the presence myosin-II motors and/or Arp2/3 complexes can enforce the cortex to undergo self-organization resulting in pattern formation. For these reasons, it is obvious to hypothesize that reduction of Arp2/3 complex activity or changes in myosin-II contractility may result in ultra-structural rearrangements of the cortex. Yet, the self-organized actin patterns nucleated by the Arp2/3 complex and/or crosslinked by myosin are expected to be fundamentally different. In the scenario of Arp2/3 complex forming, for example, asters, the nucleators would localize to the aster centres and nucleate new F-actin from their pointed (−) ends leaving their barbed (+) ends pointing outwards. In contrast, myosin-II motor proteins walks towards the barbed (+) ends of F-actin and hence would accumulate the barbed (+) ends at the aster centres (Fig. 1a).

To experimentally confirm the theoretical considerations and to identify the microscopic processes underpinning the self-organization of cytoskeletal organization, various in vitro model systems of purified myosin-II motors and filaments have been developed. Similar to the theoretical predictions, these membrane-free systems revealed that actin self-organization such as aster or star formation occurs through an active multistage coarsening process[16], in which myosin-II motors form dense foci by moving along the actin network structure and which is followed by coalescence and foci accumulation of actin filaments in a shell around them. Consequently, myosin actively reorganizes actin into the variety of mesoscopic patterns but only in the presence of bundling proteins such as fascin[17]. Above a critical concentration or activity, these motors can also prevent polymerization and bundling of actin filaments[17] and shift its organization between states, for example, from vortices to asters[18,19]. Such non-equilibrium-driven system can exhibit a large variety of self-organized structures that not only include patterning such as asters and vortices but also contractile or oscillatory states, nematic order, depending on the underlying class of microscopic processes, of which some have been reported[3,20–23].

Conclusive evidence from computational analysis and actin gel assays in vitro have therefore demonstrated how in membrane-free systems collective action of purified myosin-II motors or of the Arp2/3 complex organizes F-actin into patterns[13,24,25]. If also emerging in cells, these self-organized actin structures may have considerable effects on cellular mechanics and membrane organization, which might further differ between the different patterns[14]. Computer simulations have recently outlined the physiological importance of the actin pattern organization by highlighting their direct implications onto the molecular architecture and dynamics of the plasma membrane. Specifically, it has been predicted that actin aster formation can have a direct impact on the clustering of membrane-associated molecules[26]. Similarly, it has on several occasions been experimentally demonstrated how cortical actin organization may influence molecular plasma membrane dynamics[27–33]; however, direct evidence of the predicted actin self-organization dynamics in the living cell remains unknown. This is partly due to the complexity of the system that is tightly regulated by cellular signaling processes and partly due to the limitations of the employed optical imaging modes that precluded the direct observation of the microscopic

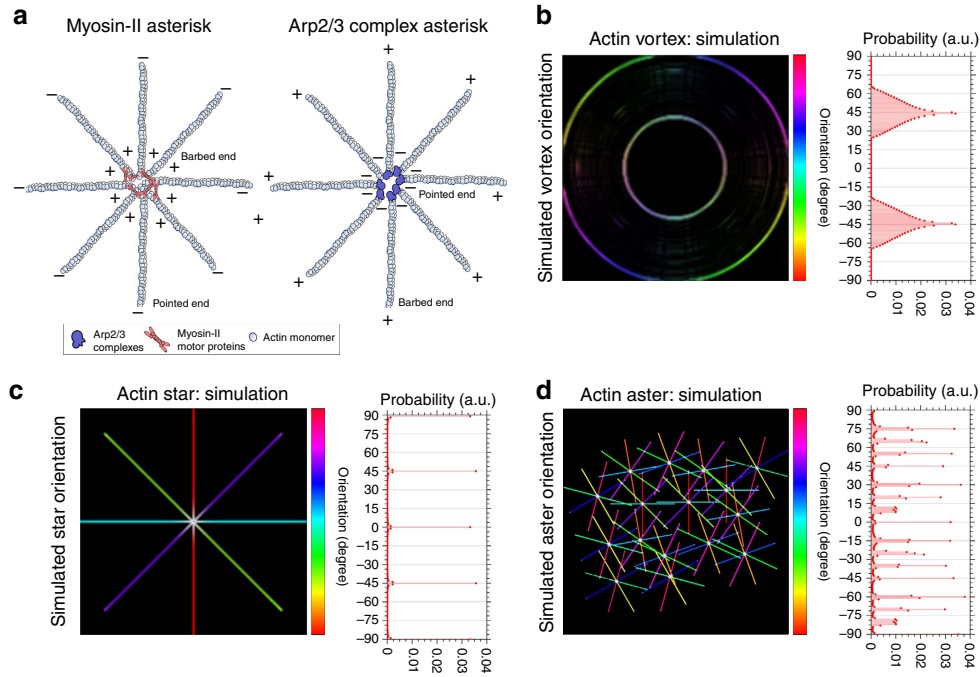

**Figure 1 | Schematic of the self-organization of the actin cortex.** (**a**) Two main nucleation pathways of actin patterning: (right) myosin II motor proteins crosslink F-actin (out of actin monomers) at their barbed ends ($+$) at the pattern centres resulting in the point ends ($-$) pointing outwards; and (left) Arp2/3 complexes bind to preexisting F-actin and nucleate new filaments from their pointed ends ($-$) leaving the barbed ends ($+$) pointing outwards. (**b**–**d**) Simulations of self-organizing actin patterns as proposed from *in vitro* data: vortices (**b**), asters (**c**), and stars (**d**). Left panels: principal spatial organization of actin with colour scale highlighting orientation of F-actin arms in space (light blue horizontal to red vertical, $-90$ to $90$). Right panels: Frequency histogram of spatial orientations of F-actin arms (given as probability distribution), highlighting characteristic distributions for each case.

pattern formation of the cellular actin cortex owing to missing spatial resolution[34,35]. In the living cell, for instance, myosin motor activity is usually tightly regulated; otherwise an initially well-connected network would evolve into a critical state where ruptures form across the entire network[36]. Also, the referenced prior *in vitro* and *in silico* studies used very specific components and assumptions, which are not necessarily identical to the mechanisms in the cell cortex.

Using advanced microscopy, we here show the self-organization of the cortical actin into vortices, stars and asters in HeLa cells. Unexpectedly, these processes are primarily driven by the Arp2/3 complex but not myosin. Scanning electron microscopy (SEM) on fixed cells reveals that actin stars form throughout the cortex of adherent cells, including the apical membrane side. Blocking Arp2/3 complex nucleation enforces cortex reorganization into small asters rather than larger stars as predicted from theory. Further, super-resolution optical stimulated emission depletion (STED) microscopy and a combination of high-resolution optical extended total internal reflection fluorescence and structured illumination microscopy (eTIRF-SIM) allows the monitoring of these transitions over time in living cells, which demonstrates that upon adherence of the cells an active multistage coarsening process naturally leads to the formation of actin vortices and subsequently into stars and asters. Measurements of cell mechanics and plasma membrane order or fluidity indicate that patterning alters cellular membrane architecture but occurs at constant cortical elasticity. Tuning the activity of the Arp2/3 complex to alter filament assembly may thus be a mechanism allowing cells to adjust their membrane architecture without affecting their macroscopic mechanical properties.

## Results

**Self-organizing actin patterns.** To assess the self-organized pattern organization of cortical actin, we used a combination of experiments and computational analysis. We first simulated the distributions of F-actin orientations and densities within actin vortices, stars and asters. Qualitative analysis of the simulated distributions allowed us to identify and differentiate these actin patterns from randomly generated F-actin networks in the experiments. To shed light onto the mechanisms that lead to the formation of these patterns, the architecture and spatio-temporal dynamics of the actin structures were then monitored at three time points (2, 4 and 12 h) from the first contact of the cells with the glass substrate until full substrate adherence during cortex assembly and homeostasis. Further, we studied the effects of Arp2/3 and myosin-II inhibiting reagents on these actin structures; here we always introduced dimethylsulfoxide (DMSO) controls (without the drugs), as the drug treatments were performed in buffers containing DMSO. Finally, we investigated how pattern organization affected the architecture of the plasma membrane and cellular mechanics. The choice of methods and their advantages and limitations are further outlined in Supplementary Note 1.

We first set out to define the different actin patterns such as vortices, asters and stars in greater detail. As further outlined in the Introduction section and Supplementary Note 2, we distinguish between vortices as ring-like structures with high rotational symmetry and asters and stars as patterns with asterisk-like topology around a clear nucleation centre and radiating F-actin arms (Fig. 1b). We differentiate between the latter two by defining asters as patterns with a maximum of $< 3 \mu m$ in diameter and arms shaped by individual F-actin fibres and stars by an overall larger size with diameters of multiple micrometres shaped by F-actin bundles. The diameter of asters and stars was calculated by measuring the mean absolute value of the width of each F-actin bundle radiating outwards, starting at the pattern core to both opposing ends of the F-actin bundles. To formalize these criteria, we simulated the pattern geometry and

computed the corresponding distribution of fibre orientations within vortices, stars and asters (Fig. 1b–d). Typically, vortices displayed a frequency distribution of two distinct bell-shaped peaks at ± 45° owing to their finite thickness (Fig. 1b). For stars, the frequency distribution of the fibre orientations showed a few distinct peaks characteristic for their arm orientations (Fig. 1c), while for asters it consisted of a series of peaks owing to the many individual fibres radiating from their centres (Fig. 1d). In contrast, randomly structured F-actin networks (whether bundled or individual fibres) showed a uniform distribution of fibre orientations (Supplementary Fig. 1).

**Actin vortices in living cells**. To highlight the potential emergence of aforementioned self-organized actin patterns in cells, we investigated the organization of the actin cytoskeleton at the basal membrane of live HeLa cells at early stages of cell adherence. We let the HeLa cells adhere to the plain glass substrate for around 2–3 h and then started our experiments (see Methods section). We chose this cell line because the proteic composition and ultrastructural organization of its cortex is well characterized[6,10,37,38]. To study the microscopic dynamics underpinning pattern organization in living cells, we used optical eTIRF-SIM microscopy (see Methods section)[39], which allowed fluorescence recordings of molecular cell dynamics at extended spatial (90-nm lateral spatial resolution) and high temporal (one image frame per second) resolution in a 150–200-nm-thick layer above the coverslide, such as the basal actin cortex. We first studied the dynamics of stably expressed fluorescently tagged F-actin (Lifeact-citrine) at these conditions. Figure 2a shows snapshots and temporal projections (TPs) of the spatial organization and distribution of F-actin. We were able to record images over a period of 90–300 s (Supplementary Fig. 2 and Supplementary Movies 1 and 2). We discovered ring-like actin structures with characteristics of vortices (Fig. 2a and Supplementary Movie 3). The distribution of fibre orientation for these structures was reminiscent of the simulated distributions of orientations (Fig. 2a,b). The vortex-like structures showed a high degree of activity, with new actin filaments continuously emerging from their outer ring (Fig. 2c and Supplementary Movie 4) as well as some degree of rotation around their centre (Fig. 2c and Supplementary Movie 5). During their appearance, the vortices retained a well-conserved diameter of 500 ± 50 nm (s.e.m., N > 120) (Fig. 2d). Notably, vortices occurred to be free without any interconnections to peripheral actin.

**Nucleation and maintenance of vortices by Arp2/3**. Next we investigated the dynamics of the Arp2/3 complex in relation to those of the actin vortices using eTIRF-SIM on live HeLa cells (incubated for 2–3 h as before, and expressing Lifeact-citrine as well as the halo-tagged p16 sub-unit of the Arp2/3 complex[40] labelled with the membrane-permeable organic dye JF542 (ref. 41)). The Arp2/3 complexes continuously attached to and detached from F-actin, indicating the importance of the Arp2/3 complex in vortex nucleation and maintenance at this stage (Fig. 2e and Supplementary Movie 5). We then examined the responses of these structures to disturbances of the actin cortex by blocking the branching and growth of new filaments using the well-characterized Arp2/3 complex inhibitor CK666 (see Methods section). After addition of CK666, the Arp2/3 complexes became immobile and thus likely stopped nucleation of F-actin. No such effects were observed in the control experiments with DMSO only. Specifically, vortices were less dynamic, for instance, the Arp2/3 complexes accumulated at the vortex periphery and at the surrounding cortical actin, showing no observable movement

during acquisition (Fig. 2f). Yet, the vortices maintained their characteristic structures, including their retained well-conserved diameter of 500 ± 50 nm (s.e.m., N > 120, P < 0.01 (Student's t-test) compared with wild-type and DMSO control conditions, Fig. 2d).

Moreover, we determined the involvement of myosin-II motor proteins in the nucleation and maintenance of actin vortices. eTIRF-SIM measurements of untreated HeLa cells stably expressing fluorescent F-actin (Lifeact-citrine) and myosin-II (halo-tagged myosin regulatory light chain 2 (MRLC2) labelled with the membrane-permeable organic dye JF542) showed that myosin patches localized only to the surrounding F-actin bundles but not to actin vortices, with an active mobility along the filaments (Fig. 2g), which is in contrast to the aforementioned characteristic distribution of the Arp2/3 complex at the centre of actin vortices (Fig. 2i). In order to test whether myosin motors were essential for the maintenance of vortices, we pharmacologically blocked MRLC phosphorylation by the Rock/Rho-kinase inhibitor Y27632 (see Methods section). Yet, the cortical actin did not show any difference in the organization of the vortices on the total time scale of the eTIRF-SIM measurements, and vortices maintained their characteristic structures and dynamics (Fig. 2h), including their retained well-conserved diameter of 500 ± 50 nm (s.e.m., N > 120, P < 0.01 (Student's t-test) compared with wild-type and DMSO conditions, Fig. 2d).

**Transition from vortices into asterisk-like patterns**. At later time points (< 4-h incubation time), we observed the transition of actin vortices into asterisk-like structures. Figure 2j displays a time course of actin vortices transforming into asterisk-shaped patterns (see also Supplementary Movie 6). At the periphery of the vortices, multiple actin filament strands were generated and then the actin structures interconnected to the surrounding cortical actin within a short period of time (< 10 s), urging on the transformations. These observations highlighted that self-organized pattern formation is fast and that vortices form prior to actin asterisk-shaped patterns.

**SEM of actin patterns in fixed cells**. To investigate the asterisk-shaped actin structures in more detail, we first imaged the surface topography of cervical HeLa cells at the highest possible spatial resolution using SEM (see Methods section). We let the cells adhere for around 4 h and then fixed them using SEM-dedicated protocols (see Methods section). Figure 3a shows representative SEM images of the apical side of the HeLa cells, which accurately highlight nano-structural details of the F-actin network. We found large asterisk-like features with an F-actin focus in the centre (red arrow) and outward radiating filaments (white arrows). Inhibition of the activity of the Arp2/3 complexes using CK666 revealed the appearance of more densely arranged smaller asterisk-like features (Fig. 3b; red arrows) that are all interconnected to one another through actin filaments. The average total diameter (core and arms) of the larger actin structures (without treatment) was $\delta = 5.0 \pm 1.1\,\mu m$ (s.e.m., N = 58), while $\delta = 2.4 \pm 0.4\,\mu m$ (s.e.m., N = 98) of the smaller structures (with treatment) was on average half the size (Fig. 3c). No transition effects were observed in the control experiments with DMSO only. Further, the two structures revealed very different nanoscale organizations (Supplementary Table 1). The arms of the larger patterns predominantly consisted of multiple F-actin bundles (white arrow at the outermost right panel, Fig. 3a) while the arms of the smaller comprised mostly single bundles or individual actin filaments (white arrow at the outermost right panel, Fig. 3b). Filament bundles in the larger structures displayed distinct orientations (Fig. 3d) while those of the individual actin filaments

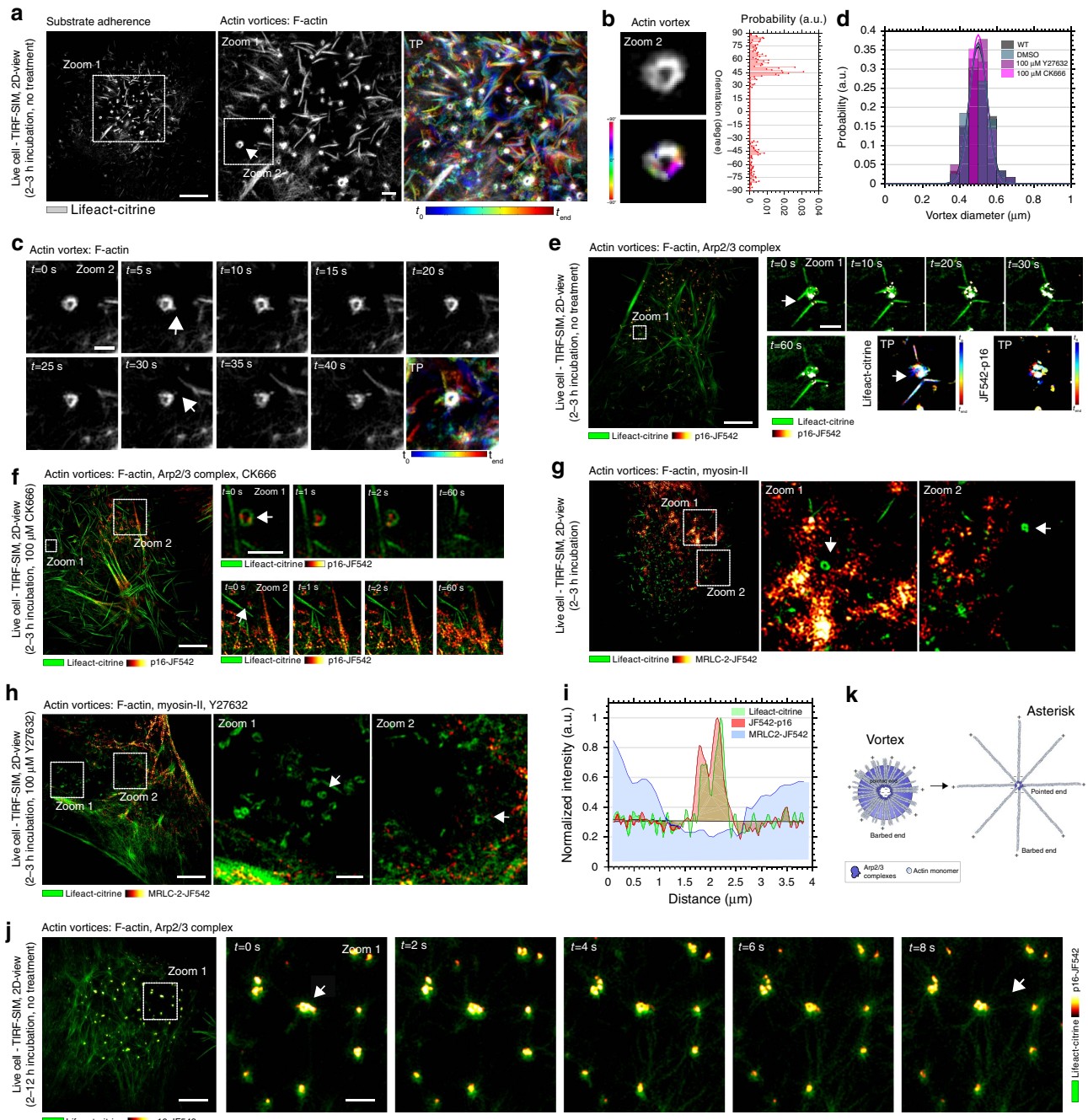

**Figure 2 | Actin vortices.** Representative eTIRF-SIM images of the dynamics of actin (Lifeact-citrine), Arp2/3 complex (p16-JF542) and myosin-II (MRLC2-JF542) at the basal membrane of live HeLa cells after short (2–3 h) incubation times. (**a**) Actin (grey scale) at one time point (left: overview; middle: zoom-in into white box, arrow: vortex) and TP (right, zoom-in as in middle) over the 60-s time course of the recording (as labelled, visible and white colours indicate mobile and immobile features, respectively). (**b**) Zoom-in on individual vortex (upper left, grey scale) and respective spatial orientation map of F-actin (lower left, light blue horizontal to red vertical, − 90 to 90) with (right) frequency histogram of orientations (probability distribution). (**c**) Zoom-in on individual vortex (white box in middle **a**) at various time points (as labelled) and temporal projection (TP, as in **a**). (**d**) Histogram of vortex diameters (probability distribution, $N = 120$ vortices from 12 cells) for different conditions as labelled. (**e,f**) Actin vortices (green, arrows) and Arp2/3 (red to yellow): overview (left) and zoom-ins (into white box) at various time points (as labelled) without (**e**) and with CK666 treatment (**f**). (**g,h**) Actin vortices (green, arrows) and myosin II (red to yellow): overview (left) and two zoom-ins (into white boxes) without (**g**) and with Y27632 treatment (**h**). (**i**) Representative intensity line profile through a vortex for actin (green), Arp2/3 complex (red) and myosin II (blue). (**j**) Transformation (arrows) from vortices to stars: Actin (green) and Arp2/3 (red to yellow) shortly after 3-h incubation (left: overview; right: zoom-ins into white box) at various time points (as labelled). (**k**) Schematic of F-actin (with barbed + and pointed − ends) and Arp2/3 complex organization for vortices (left) and asterisk-like patterns (right) with potential transfer (arrow). Scale bars: 5 μm (overviews) and 1 μm (zoom-ins).

in the smaller structures showed a continuum (Fig. 3e). Following our initial definitions (Fig. 1), we identified these structures as larger actin stars in the case of untreated and smaller asters in the case of CK666-treated cells. In addition, the F-actin bundles originate rather straight out of the star's centres, while the filaments originating from the asters buckled and thus appeared to

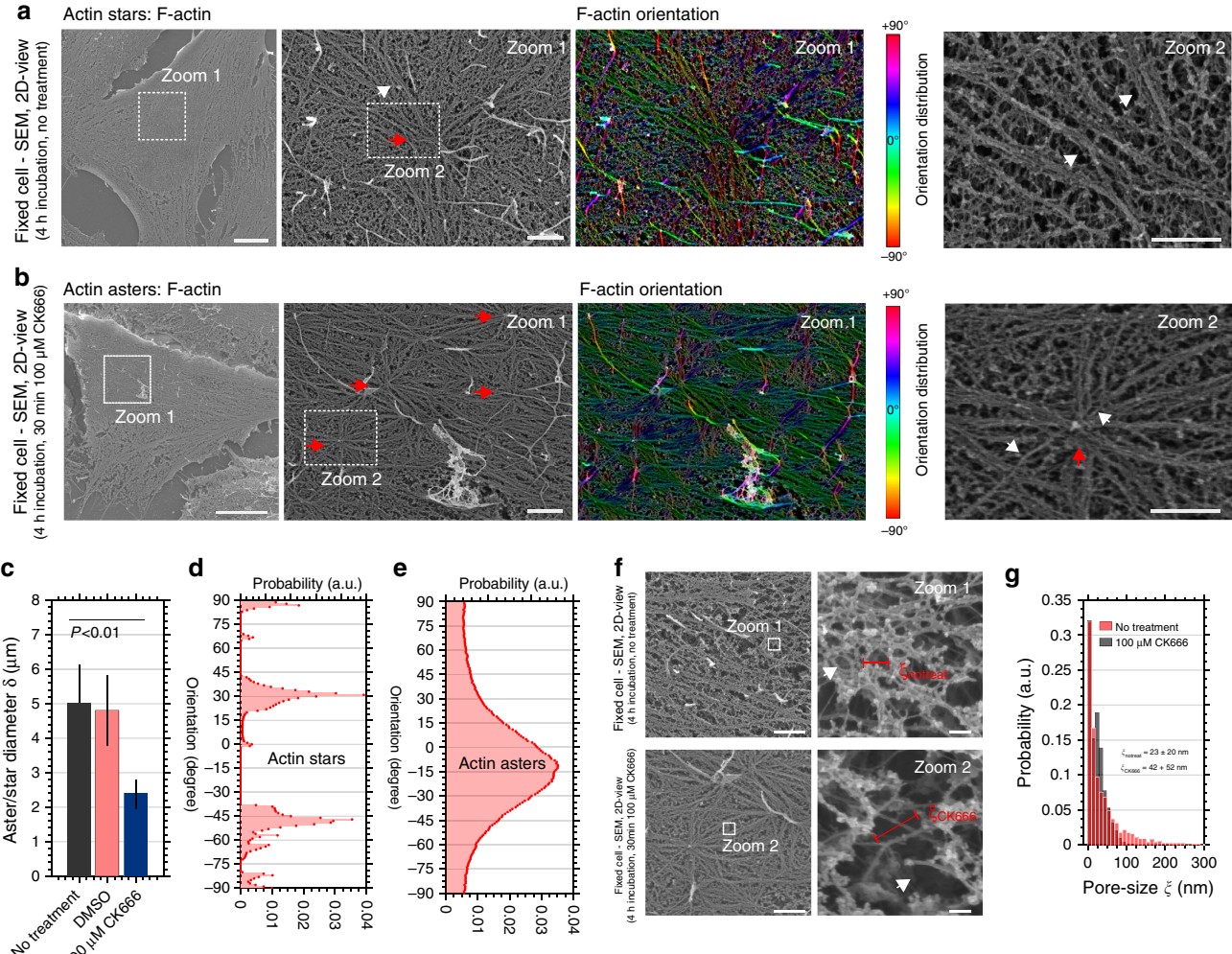

**Figure 3 | Characterization of asterisk-like actin patterns.** Representative SEM images of the apical side of fixed HeLa cells after 4-h incubation time. (**a,b**) Overview (left), zoom-in (into white box, middle left) with respective F-actin spatial orientation map (middle right, light blue horizontal to red vertical, − 90 to 90) and further zoom-in (into white boxes, right) without (**a**) and with (**b**) CK666 treatment, depicting (**a**) stars (red arrows) with radially-aligned straight filaments (white arrows) and (**b**) smaller-scale asters (red arrows) characterized by outgoing shorter and buckled filaments (white arrows). (**c**) Diameters $\delta$ (core plus outgoing filaments) of asterisk-like actin pattern without treatment (black), under DSMO control conditions (red) and after CK666 treatment (blue). Mean and error bars (s.e.m.) from $N_\delta = 58$, $N_{\delta,\ DMSO} = 39$ and $N_{\delta,\ CK666} = 98$ features on 12 SEM images from 12 different cells per samples. $P$ values from Student's $t$-test. (**d,e**) Frequency histogram of spatial orientations of F-actin arms (probability distribution) without (**d**) and with CK666 treatment (**e**), highlighting the characteristic distributions for stars and asters, respectively, as marked. (**f**) High magnification SEM (right: zoom-ins into white boxes) without (upper) and with CK666 treatment (lower) indicating changes in sizes of pores within the actin mesh (red bars, pore size diameter $\xi$). (**g**) Probability distribution of the diameters $\xi$ of the actin pores ($N_\xi = 2950$ and $N_{\xi,\ CK666} = 3560$ features on 12 SEM images from $N = 12$ different cells per samples) without (red) and with CK666 treatment (grey) showing exponential distribution with given mean values $\xi_{notreat}$ and $\xi_{CK666}$, respectively. Scale bars: 5 μm (overviews) and 1 μm (zoom-ins and high magnifications).

be under mechanical stress. SEM images taken at higher magnification further highlighted large holes within the mesh of the actin cortex with a two-fold increased pore size after the CK666 treatment (Fig. 3f,g and Supplementary Fig. 3).

**STED microscopy of stars and asters in living cells.** We next set out to confirm the SEM results in living cells, using three-dimensional (3D) optical super-resolution STED microscopy (60 nm lateral and 300 nm axial resolution), especially to exclude that star and aster formation was a consequence of or affected by the fixation procedure used in the SEM experiments. We imaged actin filaments at the basal plane of the adherent live HeLa cells, close to the microscope's coverslide to avoid potential deterioration of the performance of the 3D-STED microscope owing

to optical aberrations[42]. Labelling was performed by the stable expression of the fluorescent F-actin reporter Lifeact-citrine. Consistent with the SEM data, we found asterisk-like features in both untreated and CK666-treated HeLa cells. In untreated cells, the patterns were larger with brighter central F-actin foci and peripheral filaments (white arrow, Fig. 4a), while inhibition of Arp2/3 complex with CK666 (over 30 min) reduced their size and general brightness (Fig. 4b). Finally, we computed the distribution of fibre orientations and confirmed the respective simulated distributions for stars and asters (Fig. 4c), again properly identifying these structures.

After 12 h (instead of 4 h) prolonged adhesion times (without CK666 treatment), we observed asters instead of stars (Fig. 4d). The feature sizes were comparable with CK666-induced structures, and the distribution of fibre orientations was reminiscent of

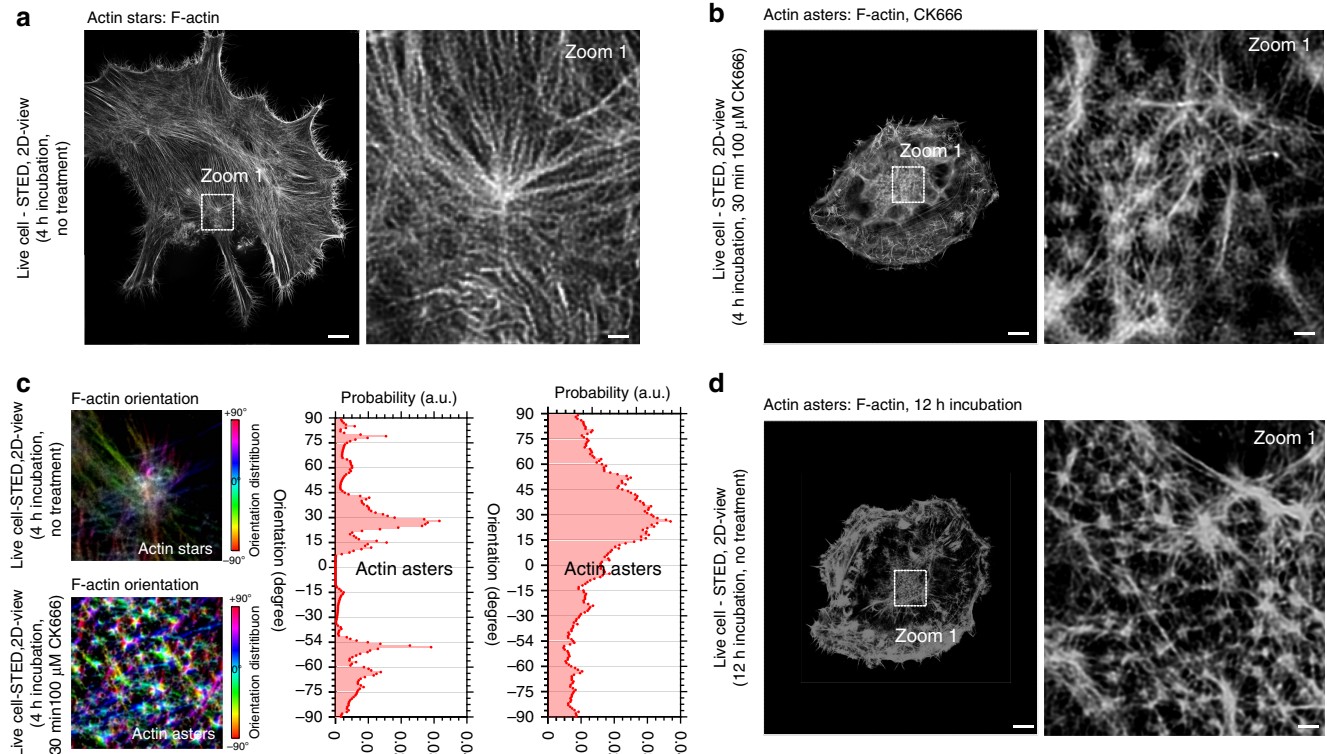

**Figure 4 | Super-resolution STED microscopy of the actin organization at the basal membrane of live HeLa cells.** (**a**,**b**,**d**) Representative overview images (left) and zoom-ins (into white boxes, right) after 4-h incubation without treatment (**a**, depicting stars), after 4-h treatment following CK666 treatment (**b**, 100 μM for 30 min, depicting asters as marked by white arrow) and without treatment but 12-h long incubation time (**d**, depicting asters). (**c**) Representative orientation map of F-actin from a region of A (upper) and B (lower, colour code: light blue horizontal to red vertical, − 90 to 90), and (right) frequency histogram of spatial orientations of F-actin arms (probability distribution) generated from the STED microscopic images without (left) and with CK666 treatment (right), highlighting the characteristic distributions for stars and asters, respectively, as marked. Scale bars: 5 μm (overviews) and 1 μm (zoom-ins).

the characteristic aster profile of CK666-induced features. Upon prolonged observations of F-actin dynamics using the eTIRF-SIM microscope, we could occasionally record the transition of stars into asters, as well as verify the transition by quantification of the appearance of the different patterns at different time points (Supplementary Note 3, Supplementary Fig. 4 and Supplementary Movies 7 and 8). Asters therefore form naturally from stars during full substrate adherence.

**Nucleation and maintenance of stars and asters by Arp2/3.** Using eTIRF-SIM, we studied the temporal dynamics of fluorescently tagged F-actin (Lifeact-citrine) and the Arp2/3 complex (halo-tagged p16, labelled with the membrane-permeable organic dye JF542) in live HeLa cells after 4-h incubation, that is, during the presence of stars (Fig. 5). The expected star-like features moved only slightly over time, and Arp2/3 complexes organized into clusters at the centres and arms of the stars (white arrow). The spatio-temporal organization of the Arp2/3 complexes was very dynamic; they continuously attached to and detached from F-actin (Supplementary Movie 9), indicating the importance of the Arp2/3 complex in star nucleation and maintenance at this stage. To further confirm this, we inhibited the activity of the Arp2/3 complexes using CK666. After 30-s CK666 treatment, the Arp2/3 complexes became immobile and thus likely stopped nucleation of F-actin at the star centres (no such effects were observed in the control experiments with DMSO). Specifically, the Arp2/3 complexes accumulated at the roots of the peripheral filaments avoiding the star core and showing no observable

movement during acquisition (white arrow, Fig. 5b and Supplementary Movies 10 and 11). Over time (30 min) CK666 treatment resulted in the aforementioned transition of actin stars into asters, where we observed a similar macroscopic immobility of Arp2/3 complex, which were now concentrated more to the aster centre (Fig. 5c and Supplementary Movie 12). These observations suggested that CK666 was indeed functional and induced the transition from actin stars to asters.

To further investigate the mechanisms underlying aster formation, we examined the dynamics of Arp2/3 complexes and asters after long incubation (12 h) without CK666 treatment, that is, for star-to-aster transformation under non-inhibited Arp2/3 activity. Similarly to the dynamics observed for actin stars, the Arp2/3 complexes were highly dynamic, with the complexes continuously attaching and detaching to and from the peripheral F-actin and the centre of the asters (Fig. 5d). The complexes now underwent continuous microscopic motion within the core, where new filaments were probably generated (or polymerized) and growing radially outwards from the centre (Supplementary Movies 13 and 14). This is in contrast to the previously mentioned arrested mobility and activity of Arp2/3 within the aster centres following CK666 treatment. This difference in mobility at the cores of the asters is further highlighted by tracking the positions of the Arp2/3 complexes within individual asters (Fig. 5e and Supplementary Movie 15). The molecules travelled within the core with a median distance $\sigma = 350 \pm 20$ nm (s.e.m., $N = 598$) from the aster centre when imaging was performed after 12-h incubation without CK666 treatment and $\sigma = 0 \pm 10$ nm (s.e.m., $N = 598$) after CK666 treatment (Fig. 5f).

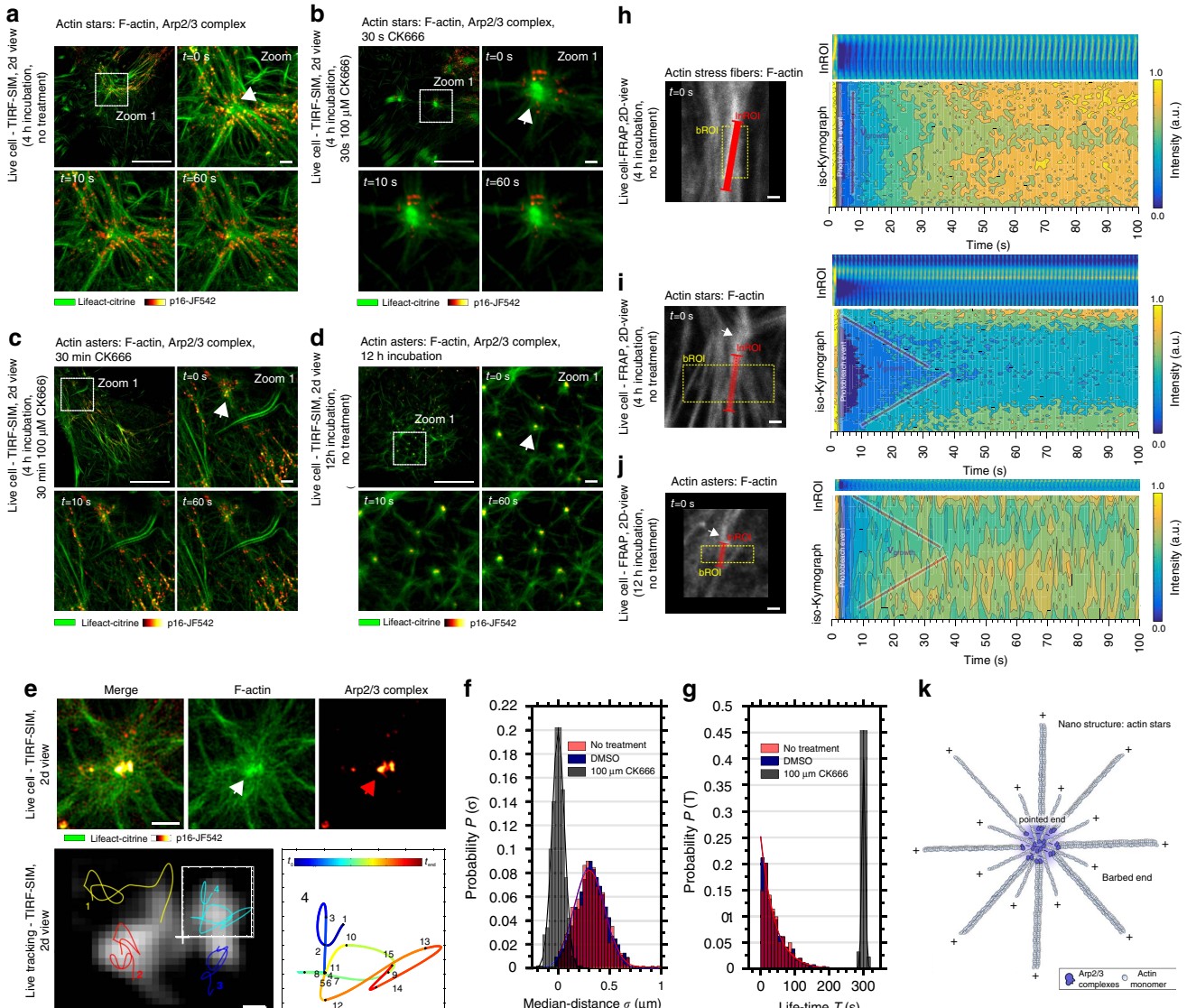

**Figure 5 | Nucleation and maintenance of stars and asters by Arp2/3.** (**a–g**) Representative eTIRF-SIM recordings of the dynamics of actin stars and asters (Lifeact-citrine, green) and Arp2/3 complexes (p16-JF542, red to yellow) at the basal membrane of living HeLa cells. (**a–d**) After 4-h incubation without treatment (**a**, stars with Arp2/3 complex mobility, white arrow), after (**b**) 30-s and (**c**) 30-min CK666 treatment (**b**: stars and **c**: asters without Arp2/3 complex mobility, white arrows) and (**d**) without treatment but 12-h long incubation time (asters with Arp2/3 complex mobility, white arrow): overview (upper left) and zoom-ins (into white boxes) at different time points (label t in the upper left corner). (**e**) Zoom-in around an individual aster (upper right Arp2/3, middle actin, left merge), respective tracks of the Arp2/3 complexes (lower left, individual tracks colour coded, overall Arp2/3 signal giving a shallow background) and detailed temporal map of the tracks within the white box (lower right, colour coding: position in individual image frames, total length 60 s, frame rate 0.5 Hz). (**f,g**) Histogram of (**f**) median distances $\sigma$ travelled in between image frames and (**g**) dwell times within the image, that is, track lengths (probability distributions, $N = 598$ tracks from 12 cells) for the Arp2/3 complexes without treatment (red), under DMSO control (blue) and after CK666 treatment (grey). (**h–j**) FRAP measurements on single actin filaments (Lifeact-citrine) for (**h**) actin stress fibres after 4-h incubation and without treatment, (**i**) fibres pointing out of stars after 4-h incubation and without treatment and (**j**) fibres pointing out of asters after 12-h incubation and without treatment: (left) representative confocal images (left, bROI and lnROI marked), (upper right) kymographs along the red lines (lnROI, intensity dark blue to yellow) and (lower right) iso-Kymographs for each pixel along the red lines (normalized intensity over time, intensity dark blue to yellow) with red lines marking directionalities in fluorescence recovery, that is, actin growth. (**k**) Interpretation of measurements with schematic of F-actin (with barbed + and pointed − ends) and Arp2/3 complex organization for actin stars. Scale bars: 5 μm (overviews) and 1 μm (zoom-ins and FRAP images), 100 nm (**e**).

Further, the residence time (or dwell time) of the Arp2/3 complexes at the asters (that is, the time between appearance and disappearance) was exponentially distributed (indicating a random process) with a mean lifetime of $\tau = 20 \pm 10$ s (s.e.m., $N = 598$) following no CK666 treatment (or under control conditions in the presence of DMSO only) and a 12-h long incubation and Gaussian-distributed about a mean $\tau = 300 \pm 5$ s (s.e.m., $N = 598$) after CK666 treatment (Fig. 5g). Note that the

latter time is identical with the total lengths of the observation and thus an indication of the immobilized Arp2/3 complexes with negligible on–off kinetics.

In contrast to the Arp2/3 complex, pharmacological treatments with the Rock/Rho kinase inhibitor Y27632 using eTIRF-SIM revealed that myosin-II was not actively participating to the nucleation and maintenance of actin stars and asters (Supplementary Note 4 and Supplementary Fig. 5).

**Polarization states of stars and asters**. In light of the importance of the Arp2/3 complex, we next aimed to further study the structural organization of actin asters and stars. As highlighted in the Introduction section, actin filaments are inherently polarized. Within actin filaments, actin protomers point from the barbed end ( + ), the location of polymerization, towards the direction of the pointed end ( − ), the location of depolymerization and where the Arp2/3 complexes are situated.

For that, we performed fluorescence recovery after photo-bleaching (FRAP) experiments on actin filaments (labelled via Lifeact-citrine) to report on their turnover dynamics and thus on the filament's directionality or polarity (see also Supplementary Note 5 and Methods section). We specifically monitored fluorescence recovery over time at a linear region of interest (lnROI) parallel to actin fibres, for example, along the star or aster arms (Fig. 5h–j). If recovery occurred only at one of the ends of the lnROI, for instance, close to the centre of the star/aster, this would suggest that the growing barbed ends of the actin fibres were located at the core of the patterns. In contrast, no obvious directionality of the recovery would exclude filament sorting within the patterns and preclude structural conclusions. To visualize any directionality in the fluorescence recovery, we computed two types of kymographs: (1) a classic kymograph depicting the intensity along all points of the lnROI over time (upper right panels, Fig. 5h–j), and (2) a so-called iso-Kymograph displaying a contour map of the fluorescence recovery in the two-dimensional space of line position and time (contour lines represent values of equally recovered intensity; lower right panel, Fig. 5h).

As a control, we first investigated the dynamics of actin stress fibres that are composed of multiple actin filaments and that are known to possess no inherent polarity[43,44]. Fluorescence recovery was homogeneous along the fibres and completed within 100 s (Fig. 5h). We found a straight contour line throughout time, confirming the non-existing inherent F-actin polarity within the stress fibres. In contrast, in both cases for actin stars and asters, we observed non-straight lines in the iso-Kymopgraphs (Fig. 5i,j), indicating F-actin polarity. Note that the asters were not induced by CK666 and that the measurements were performed after 12-h incubation. Surprisingly, recovery occurred not only from both ends of the lnROI at locations close to the star and aster centres but also from the peripheral F-actin fibres. Fitting the recovery velocity in the iso-Kymographs (red lines, Fig. 5i,j) revealed regrowth velocities of $v_{growth, star} = 60 \pm 15 \, nm \, s^{-1}$ (s.e.m., $N > 20$) in stars and $v_{growth, aster} = 58 \pm 6 \, nm \, s^{-1}$ (s.e.m., $N > 20$) in asters that yielded on average a recovery of 21–22 actin monomers per second (considering an actin monomer size of 2.7 nm). These rather slow turnover dynamics are consistent with Arp2/3 (but not formin) mediated F-actin polymerization in HeLa cells[6].

From our several observations: (i) the FRAP experiments, (ii) the previous observations that Arp2/3 actively organizes to both the centres and peripheral arms of stars and asters, and (iii) from the fact that Arp2/3 complexes can only bind to F-actin pointed ends where they facilitate actin nucleation, we can conclude that (1) nucleation of one pool of long F-actin is localized to the star and aster centres, with the F-actin's peripheral growing barbed ends pointing outwards away from the star and aster centres; and (2) an additional second pool of shorter F-actin is located at the star and aster centres with the growing F-actin barbed ends as well pointing outwards (Fig. 5k).

**Cortex mechanics and membrane architecture**. Following the nanoscale differences of stars and asters, we next investigated how actin patterning contributed to the mechanical properties of cortical actin. We characterized cortex elasticity using atomic force microscopy (AFM). For large-scale mechanical properties, the AFM cantilever was functionalized with a 10 μm bead, applying ∼0.5 μm indentation depths and micron scale (radius $r$ of about 2 μm) contact areas (Fig. 6a). Recent experimental and theoretical work indicated that under these conditions the measured stiffness is dominated by the macroscopic mechanical properties of the cortex, while mechanical contributions from the cellular cytosol are minimized[43] (Fig. 6b). We measured the elasticity under the same conditions as before, that is, on live HeLa cells attached to the microscope's cover glass. Unfortunately, for the AFM measurements, the HeLa cells have to be very firmly attached to the cover glass, which is achieved not before 4 h of incubation time. Consequently, we were not able to investigate the effects of vortex formation on elasticity but only of the actin stars and asters. Cortical elasticity was quantified by the elastic modulus of $E$ (higher values indicate a stiffer cortex). After 4 h of incubation, that is, in the presence of stars, we measured $E = 1.5 \pm 0.75 \, kPa$ (s.e.m., $N = 150$ curves, $n = 35$ cells). This value did not change significantly ($E = 1.35 \pm 0.36 \, kPa$ s.e.m., $P = 0.02$ (Student's $t$-test), $N = 160$ curves, $n = 40$ cells) after treatment with CK666 for Arp2/3 inhibition and thus after the formation of asters (Fig. 6c,d). This independence on elasticity indicates the same macroscale cortical elasticity irrespective of the transformation from stars to asters as well as the presence of asters within the cortex. Note that this is even the case despite the fact that Arp2/3 inhibition with 100 μM CK666 leads to a ∼30% reduction of cortical actin protomers incorporated in filaments, as highlighted by previous work[10]. In contrast, inhibition of myosin-II with 100 μM Y27632 or blebbistatin resulted in a significant decrease (∼33%) of the elasticity ($E = 1.01 \pm 0.34 \, kPa$ s.e.m., $P < 0.001$ (Student's $t$-test), $n = 170$ curves, $n = 45$ cells, Fig. 6c,d). When we completely blocked the cortical actin turnover by treatment with 1 μM Latrunculin B, we also observed a marked (∼66%) decrease of cortical elasticity ($E = 0.51 \pm 0.28 \, kPa$ s.e.m., $N = 145$ curves, $n = 35$ cells, Fig. 6c,d). Control experiments with DMSO were consistent with the experiments at non-treated control conditions. Taken together, our results suggest that the self-organization of the actin cortex is functionally independent from its macroscale mechanical properties.

In contrast to the independence on cortex mechanics, investigation of the membrane lipid order indicated that microscopic patterns in the form of asters alter the architecture of the immediate environment around the asters, specifically increasing the molecular lipid order (Supplementary Figs 6 and 7). These effects were not observed in the case for the actin stars (Supplementary Note 6).

## Discussion

Using advanced electron and optical microscopy in live and fixed HeLa cells, we have highlighted the occurrence of self-organized pattern organization in the eukaryotic actin cortex. Our data suggest that upon adhesion the cortex in cells undergoes an active multistage process that naturally leads to the sequential formation of actin vortices, stars and asters. As further outlined in Supplementary Discussions, such cellular self-organization is to some degree robust towards environmental changes, and it is likely that the actin cortex of the HeLa cells undergoes such a multitude of cortical actin rearrangements owing to the changes in shape in response to the adhesion processes. Further, the observed transformations are in contrast to the transitions predicted by Monte Carlo simulations and *in vitro* experiments (Supplementary Discussions). Hence, great care has to be taken when directly transferring results from *in silico* and *in vitro* experiments to the cellular conditions, and future *in silico* and

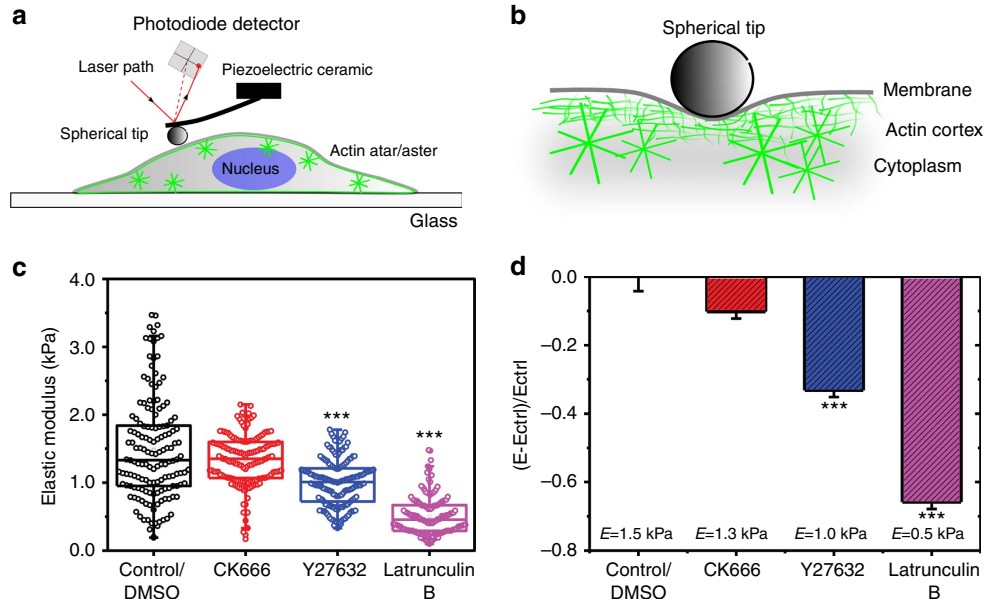

**Figure 6 | Measurements of macroscale mechanics using AFM. (a,b)** Sketch of the principal measurement concept. (**a**) The AFM cantilever coming out of the piezoelectric ceramic was functionalized with a 10 μm bead (spherical tip), and the cantilever's response measured (using the deflection of a laser detected on a photodiode) when applying $\sim 0.5\,\mu m$ indentation depths on micron scale (radius $r \approx 2\,\mu m$) contact areas on the adherent HeLa cells (including the self-organized actin cortex with asters and stars). (**b**) This experimental procedure allowed us to selectively characterize actin cortex elasticity, specifically on top of stars or asters, as under these conditions the measured stiffness is dominated by the macroscopic mechanical properties of the cortex and contributions from the cellular cytosol are negligible. (**c**) Elastic modulus $E$ measured for HeLa cells under control conditions (4-h incubation, that is, presence of stars) and after treatment with CK666 (induction of asters), Y27632 (inhibition of myosin-II activity) and Latrunculin (complete blocking of the cortical actin turnover) and (**d**) relative changes of $E$ relative to the value $E_{ctrl}$ of the control conditions, $(E - E_{ctrl})/E_{ctrl}$. ***$P < 0.001$ (Student's $t$-test). Error bars are s.e.m. from at least $N = 150$ measurements.

*in vitro* experimental conditions have to be carefully designed to match those.

In our experiments, stress response of the actin cortex and thus pattern formation followed from nucleation activity of the Arp2/3 complex (and not from increased myosin motor concentration and activity as in the *in silico* and *in vitro* experiments) and either blocking of Arp2/3 activity by the drug CK666 or adhesion processes over long (12 h) periods of time transferred stars into asters. Specifically, the eTIRF-SIM data revealed that the Arp2/3 complex localized predominantly at the cores of the stars and asters, whereas myosin-II was absent at those pattern centres, indicating that the nucleation and maintenance of actin stars and asters were independent of myosin activity. Similarly to the vortices, myosin-II may still contribute to the initiation of the pattern formation by setting the mechanical environment for self-organized pattern formation. Surprisingly, the CK666 treatments resulted not in the detachment but in the accumulation of Arp2/3 complexes in the further periphery of the actin star and aster cores. One may speculate that the Arp2/3 complexes accumulated at the pointed ends of the filament strands, that is, situated at the periphery of the pattern cores rather than in the centre of the pattern cores.

By combining these Arp2/3 characteristics with the results of our FRAP measurements on the F-actin turnover dynamics, we could extend our experimental model of the structural organization of actin stars and asters. These experiments revealed that the cores of actin stars and asters consisted of shorter Arp2/3 nucleated F-actin radiating outwards similarly to their long peripheral F-actin strands. The long actin strands radiating from the star and aster cores were multiple micrometre in size, which indicates that formin-nucleated F-actin may contribute to the formation of the pattern arms; previous studies in HeLa cells

estimated the length of Arp2/3 nucleated F-actin to a few hundreds of nanometre rather than multiple micrometres[6]. In contrast, the now determined relatively slow actin turnover velocities disclosed that both F-actin pools at the cores and peripheral strands of the stars and asters were Arp2/3 nucleated. This may potentially explain the enhanced localization of myosin-II to the long peripheral F-actin strands of especially the stars, as myosin-II would assist parallel F-actin bundle crosslinking of multiple Arp2/3 nucleated filaments.

It is not surprising that the actin cortex employs self-organization mechanisms. Cellular structures are characterized by two apparently contradictory properties: they must be architecturally stable but at the same time be flexible and prepared to respond to environmental cues. Self-organization of the actin cortex ensures structural stability without loss of plasticity. Whereas fluctuations in the components assembly dynamics do not have deleterious effects on the structure as a whole, global and persistent changes can rapidly result in morphological changes[4]. Consistent with these considerations, cortex mechanical properties were not affected by the observed actin pattern formations. Especially, the macroscale cellular mechanics as measured using AFM did not change significantly upon blocking nucleation activity of the Arp2/3 complexes and thus transformation of actin stars into asters. Cellular cortex mechanics was only altered in response to the inhibition of myosin-II-regulated contractility with Y27632 or blebbistatin or in response to deletion of the actin cortex with latrunculin B. The actin cortex and myosin contractility are the main determinants of cell mechanics, and the observed responses to Y27632, blebbistatin or latrunculin B are consistent with previous observations[6]. Myosin-II may have a minor role in the nucleation and maintenance of self-organized actin patterns to

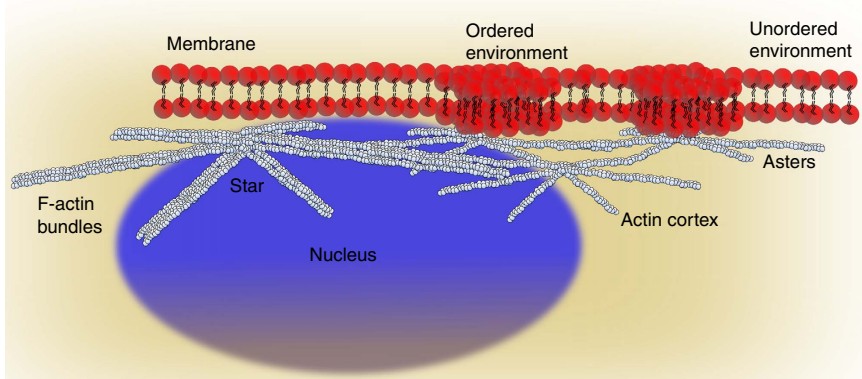

**Figure 7 | Interpretation of measurements with sketch of the actin cortex and bundles with stars and asters as well as nucleus as highlighted.** Actin patterns, specifically asters, reorganize the nearby plasma membrane environment by increasing its lipid order compared with other areas.

avoid changes in the mechanical properties of the cells. In this way, cells could effectively adjust their actin networks without alternating their mechanics.

Pattern organization did, however, affect the architecture of the cell membrane. Specifically, the plasma membrane lipid order was increased at the asters (or membrane fluidity deceased) but was not altered near the stars or actin bundles (Fig. 7). This difference in influence on the membrane organization between asters and stars may result from a tighter membrane anchoring of the asters as well as from less stable conditions for ordered environments across the larger-sized stars. These results demonstrated the influence of aster organization, following from changes in Arp2/3 activity, on the regulation of membrane architecture. Modulating Arp2/3 activity in the actin cortex, without the need for self-assembly driven by myosin-II, has the advantage that cells are able to quickly adjust membrane structure and thus functionality of membrane-associated proteins to their physiological needs, without affecting cortex contractility and thus macroscale mechanics. Cell–cell interactions, for instance, rely on a robust and stable mechanical cortex–membrane interface to efficiently organize membranous protein receptors: for instance, during the formation of the immunological synapse between lymphocytes. This is consistent with previous theoretical and experimental predictions about the relationship between actin cortex and membrane protein organization[26–33].

Although we did not observe any similarities of the observed actin patterns with podosomes (Supplementary Note 7 and Supplementary Fig. 8), future work will investigate whether structures such as podosomes are formed through similar processes and whether aster formation principles are used to nucleate podosomes. Podosomes are thought to be mechano-sensitive force sensors that might assist cells in the decision processes regarding how to interact best with tissue surfaces of different mechanical properties[45–47]. In the case of lymphocytes, osteoclasts and macrophages it has been shown that this has direct implications on their functioning[46,48,49]. This most likely follows from the fact that the organization of plasma membrane molecules such as receptors is strongly influenced by the cortical actin network. The latter is, for example, the case for immune response', which is severely affected by the ability of the immune-relevant receptors to organize within the cell membrane. Especially their organization depend on the dynamics of the Arp2/3 complex and formins that control the ultra-structural properties of the cortical actin underlying the membrane. Formins may also contribute to the mechanical settings yielding to the formation and maintenance of actin patterns as the formins

FHOD1 and INF2 have been recently demonstrated to control the *de novo* formation and contractility of podosomes[50]. Hence, our results demand ongoing research on the cortex–membrane interface to investigate whether cortical self-organization leads to further physiological-relevant effects.

## Methods

**Cell culture.** HeLa cells (product 93021013, Sigma Aldrich, UK; mycoplasma tested) were cultured in sterile DMEM (Sigma Aldrich, UK) supplemented with 10% FCS (Sigma Aldrich), 2 mM L-Glutamine (Sigma Aldrich) and 1% Penicillin–Streptomycin–Neomycin solution (Sigma Aldrich). Cells were maintained at 37 °C and 5% $CO_2$ during culturing, and handling was performed in HEPA-filtered microbiological safety cabinets.

**Plasmids.** To obtain vectors C-terminally tagged encoding Lifeact-citrine (New England Biolabs, UK), p16-halo and MRLC2-halo, the genes were amplified by PCR to produce dsDNA fragments encoding Lifeact, p16 and MRLC2 sequences, respectively, followed by a Gly-Ser linker and flanked by 5′ MluI and 3′ BamHI restriction nuclease sites. Following digestion with MluI and BamHI, this was ligated into pHR-SIN lentiviral expression vectors containing the mCitrine gene downstream of the BamHI site in the correct reading frame. Sequence integrity was confirmed by reversible terminator base sequencing. The plasmids mRuby-Vinculin (https://www.addgene.org/55885/) and pmCherry-Paxillin (www.addgene.org/50526/) were ordered from the addgene library and then amplified for the purpose of the experiments.

**Generation of stable cell lines.** HeLa lines stably expressing Lifeact-citrine, p16-halo and MRLC2-halo were generated using a lentiviral transduction strategy. HEK-293T cells were plated in six-well plates at $3 \times 10^5$ cells ml$^{-1}$, 2 ml per well in DMEM (Sigma Aldrich) + 10% FCS. Cells were incubated at 37 °C and 5% $CO_2$ for 24 h before transfection with 0.5 μg per well each of the lentiviral packaging vectors p8.91 and pMD.G and the relevant pHR-SIN lentiviral expression vector using GeneJuice (Merck Millipore, UK) as per the manufacturer's instructions. Forty-eight hours after transfection, the cell supernatant was harvested and filtered using a 0.45-μm Millex-GP syringe filter unit to remove detached HEK-293T cells. In all, 3 ml of this virus-containing medium was added to $1.5 \times 10^6$ HeLa cells in 3 ml supplemented DMEM medium. After 48 h, cells were moved into 10 ml supplemented DMEM and passaged as normal.

**Microscope coverslide preparation.** Microscope coverslide (25 mm diameter, Scientific Laboratory Supplies LTC, Germany) were washed $3 \times 1$ ml PBS at room temperature and hydrated with Leibovitz's L-15 imaging medium (ThermoFisher, UK) before use.

**High numerical aperture (high-NA) eTIRF-SIM microscopy.** High-NA eTIRF-SIM system has been described in detail[39]. Both 488-nm (Coherent, SAPPHIRE 488–500) and 560-nm laser (MPB Communications, 2RU-VFL-P-1000-560-B1R) were combined and passed through an acousto-optic tunable filter (AOTF, AA Quanta Tech, AOTFnC-400.650-TN). The beam is then expanded and sent into a phase-only modulator, which consisted of a polarization beam splitter, an achromatic half-wave plate (Bolder Vision Optik, BVO AHWP3), and a ferroelectric spatial light modulator (SLM; Forth Dimension Displays,

SXGA-3DM). Light diffracted by the grating pattern displayed on SLM passes through a polarization rotator, composed of a liquid crystal variable retarder (LC, Meadowlark, SWIFT) and an achromatic quarter-wave plate (Bolder Vision Optik, BVO AQWP3), which can rotate the linear polarization orientation of the diffracted light for different wavelengths to maintain the s-polarization, thereby maximizing the pattern contrast for all pattern orientation. The desired light of $\pm 1$ diffraction order was purified from all other high-order diffraction light by a hollow barrel mask driven by a galvanometric scanner (Cambridge Technology, 6230HB), and then they were imaged onto the back focal plane of the high-NA objective (Olympus Plan-Apochromat $\times 100$ Oil-HI 1.57NA) as two spots at opposite sides of the pupil. After collimated by the objective, the two beams interfered at the interface between coverslip and the sample at an intersection angle larger than the critical angle for total internal reflection. The generated evanescent standing wave of excitation intensity axially penetrated $\sim 100$ nm into the sample and was laterally modulated as a sinusoidal pattern that was a low-pass filtered and demagnified image of the grating pattern displayed on the SLM. The resulting fluorescence was collected by the same objective and separated from excitation light by a dichroic mirror and finally imaged onto a sCMOS camera (Hamamatsu, Orca Flash 4.0 v2 sCMOS), where the structured fluorescence raw data were recorded. The cell samples were imaged inside a micro-incubator (H301, okolabs, Naples, Italy) maintaining the physiology conditions of 37 °C and 5% $CO_2$.

For each time point, 3 raw images were acquired at successive phase steps of 0, 1/3 and 2/3 of a period of the sinusoidal illumination pattern. This process was then repeated with the sinusoidal excitation pattern rotated by $+120°$ or $-120°$ with respect to the first orientation. As the excitation pattern was conjugated to the grating image displayed on SLM, the phase stepping and pattern rotation could be accomplished by translating and rotating the grating image accordingly. A total of nine raw images were acquired for a single excitation wavelength before being switched to the next and then this acquisition procedure was repeated for each time point. Finally, the raw images were processed and reconstructed into SIM images by a previously described algorithm[51]. eTIRF-SIM experiments are presented in Figs 2 and 4–6. For each experimental condition, eTRIF-SIM data were acquired in at least 200 individual cells over the course of at least 3 independent experiments.

**Drug treatment.** Pharmacological actomyosin-specific reagents Latrunculin B, CK666 and Y27632 (Merck Bio-sciences, UK) were added to the culture medium at the given concentrations and the cells were left to incubate between 30 s and 30 min, as indicated in the corresponding experiment description. Notably, inhibitors were also present at the same concentration in the imaging medium. Drug treatment experiments are presented in Figs 2–7.

**SEM microscopy.** SEM sample preparation was performed as described in refs 10,52 with minor modifications. Two hours prior to sample preparation, whole cells were plated onto 12-mm glass coverslips. Immediately prior to fixation, the coverslips were washed three times with Leibovitz's L-15 imaging medium (ThermoFisher, UK) without serum (for cells) and transferred to cytoskeleton buffer (50 mM Imidazole, 50 mM KCl, 0.5 mM $MgCl_2$, 0.1 mM EDTA, 1 mM EGTA, pH 6.8) containing 0.5% Triton-X and 0.25% glutaraldehyde for 5 min. This was followed by a second extraction with 2% Triton-X and 1% CHAPS in cytoskeleton buffer for 5 min before washing the coverslips in cytoskeleton buffer three times. The remainder of the protocol was identical to ref. 10. The cells were then dehydrated with serial ethanol dilutions, dried in a critical point dryer, coated with 5–6-nm platinum–palladium and imaged using the detector of a JEOL7401 Field Emission Scanning Electron Microscope (JEOL, Tokyo, Japan). SEM experiments are presented in Fig. 3a,b,f. SEM data were acquired in at least 20 individual cells for each experimental condition over the course of at least 3 independent experiments.

**Orientation analysis.** The simulated fibre geometries of vortices, asters and stars were computed in custom-written MATLAB routines (MATLAB Inc., UK). The orientation and isotropy properties of a given region of interest in an image were computed based on the evaluation of the structure tensor in a local neighbourhood using the Java plug-in for ImageJ (http://imagej.nih.gov/) 'OrientationJ'. OrientationJ has four functionalities: visual representation of the orientation, quantitative orientation measurement, making distribution of orientations, and Harris Corner detection, as outlined in (www.epfl.ch/demo/orientation/). Specifically, the user specifies the size of a Gaussian-shaped window, and the program computes the structure tensor for each pixel in the image by sliding the Gaussian analysis window over the entire image. The local orientation properties are computed and are then visualized as grey level or colour images with the orientation being typically encoded in the LUT colour (hue). The data presentation was performed in custom-written MATLAB routines (MATLAB Inc.). Orientation analysis is presented in Figs 1b–d,2b,3d,e and 4c and Supplementary Fig. 1. Orientation analysis was performed in at least 20 individual cells for each experimental condition.

**Asterisk diameter quantification.** To characterize the diameter of asterisk-like patterns from the SEM images, a multi-stage image-processing protocol was performed. Custom-written MATLAB routines (MATLAB Inc.) were employed to measure the diameter of the asterisk-like patterns by fitting a line from the pattern core to the outer peripheral actin strands of each individual actin pattern (see Results section). Corresponding distributions for the diameters of each pattern type such as actin stars and asters were then presented in a boxplot for the different experimental conditions (Fig. 3c). The medium diameter was then determined for each individual structure from the distributions. Asterisk diameter quantification are presented in Fig. 3c. Diameter quantifications were performed in at least 20 individual cells for each experimental condition.

**F-actin pore size.** To characterize the pore size of the actin cortex from the SEM images as shown in Fig. 3f,g, a multi-stage image-processing protocol was performed. Prior to processing, each image was normalized to its maximum intensity. Custom-written MATLAB routines (MATLAB Inc.) were employed to segment each image by applying a threshold at an arbitrary low value (0.002), resulting in a binary image of the actin mesh. The segmented images were then inverted and the pores were interpolated with ellipsoids. The medium radius (pore size) was then determined for each individual pore and for multiple cells. Corresponding distributions of all pore sizes were then plotted into a histogram for the different experimental conditions (Fig. 3g). F-actin pore size measurements are presented in Fig. 3f,g. F-actin pore size quantifications were performed in at least 15 individual cells for each experimental condition.

**Temporal projections.** To calculate the TPs from eTIRF-SIM time lapses as shown in Fig. 2a,c, a multi-stage image-processing protocol was performed. The ImageJ plugin 'Temporal color-code' (ImageJ, http://imagej.nih.gov/) was used to super-impose all images of a time lapse onto one plane coding each frame with cold to warm colours representing early to late time points. Custom-written ImageJ look up tables were employed to represent early time frames in blue and late ones in red. TPs are presented in Fig. 2a,c.

**STED microscopy.** STED experiments were performed on a Leica TCS SP8 3X microscope (Leica, Mannheim, Germany). All live-cell experiments were performed at 37 °C and 5% $CO_2$. The microscope was equipped with a pulsed super-continuum white-light laser (Koheras SuperK, 80 MHz) for excitation and a 592-nm 1.5-W CW STED laser. Citrine was excited at 488 nm, and fluorescence emission was collected at around 530 and 520 nm, respectively. For STED imaging, excitation laser intensities of 1–5% of the white-light laser and 30–100% of the 592-nm STED laser were utilized to obtain a strong enough fluorescence signal as well as sufficient improvement in spatial resolution, and images were acquired at 1–5-s intervals to minimize loss of fluorescence owing to photo-bleaching as well as cell phototoxic effects (of which we did not observe any in the recordings). All images were acquired on the Leica HyD detectors using time-gated detection with a time gate of 1.0–1.5 ns. The STED laser beam was split into two paths each including a phase plate for creating the donut-like intensity focal patterns along lateral $(xy)$ and axial $(z)$ directions, respectively. By setting the relative allocation of the STED laser power between these two phase plates by use of a variable beam splitter, it was possible to tune the spatial resolution of the microscope along the $xy$ and $z$ direction individually.

Using the Huygens STED-Deconvolution-Wizard (Huygens Software, The Netherlands), only a moderate degree of deconvolution was applied to the recorded STED images to avoid deconvolution artefacts. The microscope's point spread function was directly calculated from the Leica imaging files, following standardized Huygens software guidelines (http://www.leica-microsystems.com/science-lab/huygens-sted-deconvolution-quick-guide/). The effective size of the STED microscope's observation spot (70 and 300 nm lateral and axial diameter, respectively) was determined using a fluorescent micro-spheres (FluoSpheres, yellow–green (505/515), diameter 0.04 μm, Invitrogen, USA). This sample was prepared by diluting the beads in Milli-Q-water with dilution factor 1:10,000. A drop of diluted beads was attached to the coverslide using poly-L-lysine (Sigma Aldrich) and then the coverslip was mounted on a microscope slide and embedded in the mounting medium mowiol. STED experiments are presented in Fig. 4a–d. STED microscopy was performed in at least 40 individual cells for each experimental condition over the course of at least 3 independent experiments.

**Particle tracking.** Particle tracking as presented in Fig. 5e was performed from eTIRF-SIM time lapses using the ImageJ plugin 'Speckle Tracker J' (ImageJ, http://imagej.nih.gov/)[53]. Custom-written MATLAB routines (MATLAB Inc.) were employed to determine the medium distance $\sigma$ of the particle from the aster centre for all time points of this trajectory and statistically analysed the tracking data. First, the centre of the aster and all positions of a particle trajectory $\mathbf{r}(t)$ was determined within a Cartesian coordinate system. Then the distance $\sigma$ between the particle and the aster centre, as the absolute value of $|r(t)|$ was determined for all time points $t$. Finally, the medium distance for one trajectory was determined and then the procedure was repeated for all the particles. Corresponding distributions of all medium distances were then plotted into a histogram for the different experimental conditions (Fig. 5e–g). Particle tracking experiments are presented in Fig. 5e–g. Particle tracking was performed in at least 30 individual cells for each experimental condition over the course of at least 3 independent experiments.

**Fluorescence recovery after photo-bleaching experiments.** FRAP experiments were performed at 37 °C and 5% $CO_2$ using a 1.4 NA $\times$ 100 oil-immersion objective on a confocal fluorescence microscope (Zeiss 780, Carl Zeiss AG, Oberkochen, Germany). The fluorescent protein citrine was excited at 488 nm and fluorescence emission was registered at around 525 nm. In the FRAP experiments, a small rectangular ROI ($\sim 5 \times 10\,\mu m^2$) centred on the actin patterns was imaged and a smaller rectangular bleaching ROI ($\sim 2 \times 4\,\mu m^2$) was chosen in its centre. This choice of imaging and bleaching modes minimized the fluorescence loss owing to photo-bleaching during the recovery as well as phototoxic effects on the cell by not exposing the whole cell to light but restricting illumination and thus photo-bleaching to a section of only 0.6 μm in thickness as outlined in ref. 54. Fluorescence recovery was monitored over a lnROI parallel to the F-actin strands for actin stress fibres, stars and asters in order to detect the direction of fluorescence regrowth. Bleaching was performed by scanning the 488 nm beam operating at 100% power of the 20 mW laser. In our protocol, bleaching was realized with a single laser pulse of 2 s during which scanning was performed with a pixel dwell time of 8 μs. The recovery of fluorescence was monitored at the same scanning speed with 1–5% of the laser power over 100 frames at 0.05–1-s intervals to minimize photo-bleaching but still allow sampling at a sufficient enough speed[54]. For each recovery, two time-lapse image streams were recorded prior to the initial bleaching, which realized normalization of the fluorescence signal. To assess the loss in fluorescence during observation of the recovery (owing to photo-bleaching), we selected the simultaneously recorded fluorescence signal from a non-bleached region. In all cases, the rate of fluorescence loss due to the observation of recovery was significantly smaller than the rates of fluorescence recovery, with a characteristic time of $\sim 1,000$ s that was one order of magnitude larger than the slowest recovery timescale observed for actin. Hence, imaging-induced fluorescence loss did not significantly affect the measurements. Conventional kymographs for the FRAP experiments were generated by computing a montage of the lnROIs using the standard ImageJ plugin Montage (ImageJ, http://imagej.nih.gov/). Iso-Kymographs were computed in custom-written MATLAB routines (MATLAB Inc.) by using the standard iso-contour functions. The F-actin turnover velocities $v_{growth}$ were computed by fitting the linear recovery trends in time in the iso-Kymographs using a total pixel size of 29 nm, frame rate of 1 s per frame and a finite actin monomer size of 2.7 nm. FRAP experiments are presented in Fig. 5h–j. For each experimental condition, we acquired FRAP recovery curves and computed the F-actin turnover velocity from at least 20 individual cells over the course of at least 3 independent experiments.

**Atomic force microscopy.** AFM indentation experiments were performed with a JPK NanoWizard I (JPK Instruments) interfaced to an inverted optical microscope (IX81, Olympus). For our measurements, we used cantilevers (MLCT, Bruker) with nominal spring constant of 0.07 N m$^{-1}$ and functionalized them by gluing a bead of 10 μm diameter (Invitrogen). Analysis of force indentations was restricted to depths of 100 to 500 nm (leading to forces <200 pN) to maximize the contribution of cortex stiffness to cantilever deflection, enabling us to probe cell cortex elasticity while minimizing contributions from the cytoplasm[55]. The actual spring constant of the cantilever was determined using the thermal noise method implemented in the AFM software (JPK SPM). Before indentation, the sensitivity of the cantilever was set by measuring the slope of force–distance curves acquired on glass regions of the petri dish. With the optical microscope, the tip of the cantilever was aligned over regions above the cell nucleus, and a couple of indentation measurements were performed on the cytoplasm. Force–distance curves were acquired with an approach speed of 5 μm s$^{-1}$ until reaching the maximum set force of 6 nN. The considered range of indentation depths was comparable to the length scale of actin patterns and therefore enabled us to detect the contribution of such networks. Using the method described by Lin et al.[56], we found the contact point, and subsequently, we calculated the indentation depth (d) by subtracting the cantilever deflection (delta) from the piezo translation (z) after contact (d = z − delta). The elastic moduli were extracted from the force–distance curves by fitting the contact portion of curves to a Hertz contact model between a pyramid and an infinite half-space[57]. AFM experiments are presented in Fig. 6a–d. AFM measurements were performed in at least 35 individual cells over the course of at least 3 independent experiments.

**Lipid packing measurements.** S-Laurdan2 (SL-2; derivative of 2-dimethylamino-6-lauroylnaphthalene) was purchased at Tpprobes (Tpprobes, South Korea). HeLa cells were plated out on 25 mm microscope glass coverslides (see Materials and Methods coverslide section) at 4 or 12 h prior experimentation. At the day of experimentation, cells were incubated with S-L2 for 5 min in PBS, washed with L15 imaging medium and measured for 30 min to minimize internalization of S-L2. S-L2 was excited at 405 nm and emission was collected between 420 and 490 nm using the GaAsP detector of the Zeiss-780 (Zeiss 780, Carl Zeiss AG, Oberkochen) inverted confocal microscope (for further details on the imaging and sample preparation protocols, see refs 58,59). Fluorescence overlay analysis was performed using the ImageJ plugin 'JACoP (Just Another Colocalization Plugin)' (ImageJ, http://imagej.nih.gov/)[60]. Notably, we calculated the Squared-overlap-coefficient $r^2$ instead of the conventional-overlap coefficient to minimize effects of background and zero intensity values in one of the two channels as outlined in ref. 60. Lipid packing measurements are presented in Supplementary Figs 6 and 7. Lipid packing

measurements were performed in at least 60 individual cells for each experimental condition over the course of at least 3 independent experiments.

**Data and code availability.** The authors confirm that all relevant data and computer codes are available from the authors upon request.

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

## Acknowledgements

We thank Satya Khuon, Helen White and Sumita Ganguly for technical support and Luke D. Lavis for the kind gifts of reagents JF542. We thank the Wolfson Imaging Centre Oxford for providing microscope facility support, the Wellcome Trust (grant ref. 104924/14/Z/14), the Medical Research Council (MRC grant number MC_UU_12010/unit programmes G0902418 and MC_UU_12025), MRC/BBSRC/EPSRC (grant number MR/K01577X/1) and institutional funding from the University of Oxford (John-Fell-Fund, EP Abraham Cephalosporin Trust Fund and Wellcome Trust Institutional Strategic Support Fund). H.C.-Y. was funded by the EPSRC. E.M. is grateful for the financial support of Wellcome Trust–Massachusetts Institute of Technology Fellowship (grant WT103883). E.S. is supported by EMBO Long Term and Marie Curie Intra-European Fellowships (MEMBRANE DYNAMICS).

## Author contributions

M.F. and C.E. developed the concept and designed the experimental approach with help from D.L. and G.C. M.F. and D.L. carried out the experiments and implemented the data analysis. M.F., C.E. and G.C. wrote the article. D.L., H.C.-Y., V.T.C., E.M. and E.B. supported with samples, experiments and experimental setups. All were involved in discussion of data and manuscript editing.

## Additional information

**Competing financial interests:** The authors declare no competing financial interests.

**Publisher's note**: 

