## [Peer Review File · Nature Communications]

Reviewers' comments:

Reviewer #1 (Remarks to the Author):

In their paper "Self-organizing actin patterns shape membrane architecture but not cell mechanics" Fritzsche and colleagues study the self-organization of actin patterns as HeLa cells adhere to a substrate. They do this to test whether, in vivo, they can observe patterns similar to or different from those observed in purified systems.

The authors use complementary microscopy techniques to show the organization of actin at the cell cortex in fixed and living cells. Many of the images and movies are beautiful and striking.

By analysing the cortex at different times the authors suggest that there is a step wise process - with a transition from actin vortices into stars and asters. They then study the role of the Arp2/3 complex and myosin in this process. In their experiments, they observe that myosin is not involved in the generation of these actin structures at the cell cortex, while Arp2/3 is. Interestingly, the addition of an Arp2/3 inhibitor to samples at early times seems to generate actin structures like those seen after 12h of adhesion. The team then use atomic force microscopy to test whether the change in organisation is associated with a change in macroscopic cortex elasticity - but find that it is not. They also suggest that there is some lipid ordering that occurs.

The results are potentially interesting but the paper has some weakness: the paper jumps around in time, doesn't properly study the evolution of different actin structures (it suggests transitions between states may be quick - but never catches one), and the data in different figures are hard to compare with one another. Thus, the manuscript could be organized better for clarity. Moreover, additional measurements are required to validate their findings before we could recommend the paper for publication in Nature Communications.

Major points:

1. The results would be easier to follow if organised in time:

- observations at 2-3 hours: actin vortices
- observations at 4 hours: actin stars
- observations at 12 hours asters

Given that this is the subject of the paper, the authors should really catch the transitions. They should also test the effects of inhibitors systematically - at each time-point. Ideally, they should include a Formin inhibitor (or at least discuss the roles of formins). Also, more effort needs to be made to make clear that the distinctions between structures are real.

2. The authors need to include a graph to show the % of actin structures at each time point - to make clear that there is a transition between structures. The experiments should be repeated using other cell lines to determine if this is a general phenomenon

3. The authors should include data to confirm that the inhibitors used work.

4. Control experiments should be performed in DMSO or carrier.

5. The authors should specify the substrate they use in their experiments and why. Is this process dependent on the substrate?

6. Currently, there is little evidence to show the reader where actin nucleation and elongation likely occur in their images. The authors should measure the rate of actin elongation in structures they can observe and should determine if these elongate in a polar or non-polar fashion. Ideally, the authors would use photo-bleaching to image sites of actin monomer incorporation.

7. Material and methods could be improved with more detailed information.

Detailed comments:

SEM reveals nanoscale organization of cortical actin stars and asters

In this part, the authors use SEM to investigate the organization of the actin in HeLa cells. Figures 1C and 1F could be improved to aid clarity. Currently they are confusing and the difference between the "no treatment" and CK666 is not clear. How do the authors measure the diameter of the asters/stars? Additional information in material and methods could clarify this point. Figure 1G could be improved to show the 2 distributions more clearly.

Super-resolution STED microscopy identifies stars and asters in living cells

In this part, the authors use STED microscopy to exclude a role of the fixation procedure in the patterns formation.

- Figures would be improved if the same scale bar was used in the 3 conditions.
- Similarly, the same ROI / zoom and scale bar should be used in the different conditions to highlight the differences in the actin patterns observed.
- Specify the time in Figure 2A

Dynamic patterning is nucleated and maintained by the Arp2/3 complex

The authors use an additional technique, TIRF-SIM, to observe the dynamics of the patterning formation. They use HeLa cells expressing Lifeact and p16, a sub-domain of the Arp2/3 complex, and they image with a time-lapse of 10s.

1. It is hard to see the dynamics of the Arp2/3 on the filaments in Figure 3. Arrows in images could help and a zoom on some zones of the images.
2. Authors should explain how they define individual Arp2/3 complexes? Co-staining should be presented to show that the probe correctly identifies complex complexes.
3. It is not clear from the data that CK666 stops the actin nucleation as stated. It is not enough to state that other studies report a 30% decrease in actin nucleation.
4. Figure 3B: what is time 0? Is this the time following drug addition?
5. What happens between 30 s (Figure 3B) and 30 min (Figure 3C)? The authors should do a longer time-lapse increasing the total time of the experiment using low laser power, lower exposure time and infrequent imaging.
6. Why does the Arp2/3 complex distribution shift after addition of the CK666 inhibitor? The inhibitor acts on the Arp2/3 complex by maintaining the complex in the inactive state. How do they explain the accumulation of the Arp2/3 complex on the filaments in the actin asters under these conditions? If the organization of the Arp2/3 complex in this structures is very dynamic, as the authors state in the manuscript, we should observe a loss of Arp2/3 complex from filaments not freezing of movement.
7. In Figure 3D it is hard to see the dynamics of the Arp2/3 complex. The authors could zoom on the region of interest to make this clearer.
8. Figure 3E: the tracking could be improved. Track 1: the Arp2/3 complex isn't seen across the whole track. Why? How do they distinguish the single molecules? How precisely (over what time period and at what scale in xyz) are aster trajectories determined?
9. TP images are very confusing. The authors could replace these images with representative single trajectories of individual Arp2/3 complexes.

Pattern formation is independent from myosin

The authors investigate the role of the myosinII in the pattern formation. The figures should have the same scale bar, and the images should be better. The authors could add some trajectories to see the effect of the inhibitor and, perhaps, a longer time-lapse to see the dynamics of the myosinII over several minutes?

Rotating vortices shape cortex assembly

We recommend that the authors start the results by discussing actin organization at shorter times.

General remarks on the observations

How many cells were observed using SEM? This is important as one would expect cells to exhibit heterogeneity like that seen in the fluorescence microscopy experiments?

Since the spatial resolution is 90 nm in these experiments, perhaps the authors could try to use TIRF instead of TIRF-SIM. With a TIRF microscope one can avoid photobleaching and phototoxicity and can decrease exposure time and/or laser power.

Relation of patterns to podosomes

This part could be better described and discussed.

Discussion

The authors describe at some point the patterns observed for microtubules in vitro. In the paper they cite, there are no stars and the transition is from asters to the vortices - not the opposite as the authors describe.

Reviewer #2 (Remarks to the Author):

The authors report the existence of not so novel actin filament-based structures in HeLa cells. Specifically, asters have been evident in fluorescent phalloidin stains for years. In addition, it is completely unclear what the significance of these structures are to cellular function. There are also several major problems with the manuscript as outlined below.

- The authors need to quantify the percentage of asters and stars at the two time points. Are there absolutely no stars at later time points? Although the authors allude to the rarity of the observed structures (present in only 30% of the cells), quantification would better make the point better that there is transition across the different structures.
- The biggest pitfall of this paper is that the authors were still not able to show the transition live. Although the authors acknowledge this, why not use TIRF SIM with lower temporal resolution? If they can get 300 total frames, why not do 5 min intervals (gives ~1500 min)?
- The myosin experiment is purely descriptive and there is no quantification. The

percentage of asters and stars with myosin absent from the core should be done. The Y drug experiment is also not very informative. What about the tension generation by just binding to actin? Try NMII knockdown and then test whether you still get self-organization of actin.

- The lipid ordering experiment is also not convincing. The gray levels are not matched and the asters don't really look like asters. Being the only figure that highlights the potential importance of these structures, there should be more quantification, gray level matching.

- In all of the live cell quantifications in Figure 3, all the Arp2/3 foci are quantified, while the number of actin aster and stars are actually scarce. Maybe another quantification of Arp2/3 mobility within a star or an aster?

- The Methods section was hastily written and has more grammatical errors than the rest of the manuscript.

Detailed response to the reviewers:

Response to Reviewer 1:

Reviewer 1 Comments to Author: In their paper "Self-organizing actin patterns shape membrane architecture but not cell mechanics" Fritzsche and colleagues study the self-organization of actin patterns as HeLa cells adhere to a substrate. They do this to test whether, in vivo, they can observe patterns similar to or different from those observed in purified systems.

The authors use complementary microscopy techniques to show the organization of actin at the cell cortex in fixed and living cells. Many of the images and movies are beautiful and striking.

By analysing the cortex at different times the authors suggest that there is a step wise process - with a transition from actin vortices into stars and asters. They then study the role of the Arp2/3 complex and myosin in this process. In their experiments, they observe that myosin is not involved in the generation of these actin structures at the cell cortex, while Arp2/3 is. Interestingly, the addition of an Arp2/3 inhibitor to samples at early times seems to generate actin structures like those seen after 12h of adhesion. The team then use atomic force microscopy to test whether the change in organisation is associated with a change in macroscopic cortex elasticity - but find that it is not. They also suggest that there is some lipid ordering that occurs.

The results are potentially interesting but the paper has some weakness: the paper jumps around in time, doesn't properly study the evolution of different actin structures (it suggests transitions between states may be quick - but never catches one), and the data in different figures are hard to compare with one another. Thus, the manuscript could be organized better for clarity. Moreover, additional measurements are required to validate their findings before we could recommend the paper for publication in Nature Communications.

We thank Reviewer 1 for their very careful reading of our manuscript and their very positive assessment of our work. We now performed and added a series of new experiments and analysis to the manuscript and completely reorganized the result and discussion sections as well as most of the Figures to avoid confusions (e.g. by jumping in between time events). In addition, we added new quantifications of the data and present our inhibitor and corresponding carrier experiments more systematically. Most importantly, we added new FRAP experiments as requested by the reviewer and discovered previously unforeseen insights into the structural organization of actin stars and asters. We also added data supporting the transitions in between actin patterns as requested by the reviewer. Please find below our responses to the detailed comments of the Reviewer 1.

Detailed Comments of Reviewer 1:

Major points:

1. The results would be easier to follow if organised in time:

- observations at 2-3 hours: actin vortices
- observations at 4 hours: actin stars
- observations at 12 hours asters

Given that this is the subject of the paper, the authors should really catch the transitions. They should also test the effects of inhibitors systematically - at

each time-point. Ideally, they should include a Formin inhibitor (or at least discuss the roles of formins). Also, more effort needs to be made to make clear that the distinctions between structures are real.

We now rewrote the manuscript organized in time as suggested by the referee.

We also added new data displaying the transition from actin vortices into stars (Figure 2J) and data displaying the transition from actin stars into actin asters (Figure 7).

2. The authors need to include a graph to show the % of actin structures at each time point - to make clear that there is a transition between structures. The experiments should be repeated using other cell lines to determine if this is a general phenomenon

We now quantify the number of patterns at a given point and the total surface area occupied by the actin patterns (if present) for all three time-points. We feel that these plots presented in Figure 7D indeed strengthen our arguments and thank the reviewer for this comment (see also Reviewer 2). We feel however that careful characterization of these structures in additional cell lines is beyond the scope of this study. We expect that our manuscript will pave the way for systematic re-evaluation of the self-organization of the actin cortex and anticipate that future studies will consider the impact of these phenomena on many biological scenarios in different cell lines as suggested by the in vitro and in silico studies. We therefore would respectfully refer the reviewer to future studies from our group and others.

3. The authors should include data to confirm that the inhibitors used work.

We used a series of pharmacological inhibitors including the Arp2/3 inhibitor CK666, the Rho-kinase inhibitor Y27632, the selective myosin-II inhibitor blebbistatin, and the actin monomer sequestering drug latrunculin B in different experiments. Prior all inhibition experiments, we always ensured to use fresh inhibitors as some of those drugs become increasingly dysfunctional over time. Despite using these inhibitors on a daily basis, we regularly perform control experiments to re-convince ourselves of the functionality of the drugs including the required carrier controls.

We apologize that we did not present all controls as outlined by the reviewer. We now added the missing control experiments and/or clarified these facts in the manuscript better. Moreover, our AFM experiments demonstrated for instance significant changes in the cortical elasticity in the response to Y27632, blebbistatin, and latrunculin B, indicating the functionality of the drugs. Additionally, we reported the short- and long-term effects of CK666 on the mobility of the Arp2/3 complex (p16 sub-units), indicating the effectiveness of the drug. We now explain this better in the manuscript.

4. Control experiments should be performed in DMSO or carrier.

Please see response above.

5. The authors should specify the substrate they use in their experiments and why. Is this process dependent on the substrate?

We now added a new section for the preparation of the coverslides to the Materials and Methods section. Additionally, we explicitly describe the coverslides in the result section and discuss the material properties of the coverslides and their potential impact on actin pattern formation in the discussion.

6. Currently, there is little evidence to show the reader where actin nucleation and elongation likely occur in their images. The authors should measure the rate of actin elongation in structures they can observe and should determine if these elongate in a polar or non-polar fashion. Ideally, the authors would use photo-bleaching to image sites of actin monomer incorporation.

We thank the reviewer for this comment. We performed and added new FRAP experiments to the manuscript. Surprisingly, we found that there are additional shorter Arp2/3-nucleated F-actin strands situated at the pattern cores (in addition to the peripheral F-actin strands) pointing outwards from the centers of the stars and asters (Figure 5H-J and Figure 5K). These new data are consistent with the eTIRF-SIM data of the Arp2/3 complexes and explain the vast amount of Arp2/3 complexes locating at the center of actin stars and asters. The Arp2/3 complexes probably nucleate these new F-actin strands at the pattern cores. We also estimated the F-actin turnover velocity as requested by the referee which was consistent with speed of Arp2/3-nucleated polymerization. We think that these experiments provided new structural insights into pattern organization and significantly strengthened the manuscript.

Let us briefly comment on the FRAP experiments. FRAP experiments cannot report on the molecular binding rates (k_{on} and k_{off}) of actin protomers to actin filaments as recently demonstrated by us (Fritzsche et al, MBoC 2013; Fritzsche et al, Science Advances 2016). FRAP experiments can however report on the flow velocity (treadmilling speed) of actin filaments which we computed from the FRAP recoveries and reported in the manuscript as requested by the reviewers. In the case of the actin patterns, these experiments were challenging, because the patterns are not visible and/or easily identifiable within the many freely diffusing fluorescently tagged actin monomers. We therefore performed the experiments using Lifeact-citrine as a F-actin reporter. Lifeact's recovery dynamics are a convolution of Lifeact binding to F-actin and actin turnover flow (actin treadmilling). Lifeact binding however is fast compared to actin turnover (otherwise we would not have been able to detect the F-actin polarity) and consequently could be here successfully employed to identify the locations of the barbed ends of the F-actin strands at the patterns, the F-actin polarity (direction of fluorescence recovery), and the F-actin turnover velocity.

7. Material and methods could be improved with more detailed information.

We apologize to the reviewer. We now carefully reworked the Materials and Method section.

Detailed comments:

SEM reveals nanoscale organization of cortical actin stars and asters

In this part, the authors use SEM to investigate the organization of the actin in HeLa cells. Figures 1C and 1F could be improved to aid clarity. Currently they are confusing and the difference between the "no treatment" and CK666 is not clear. How do the authors measure the diameter of the asters/stars?

Additional information in material and methods could clarify this point. Figure 1G could be improved to show the 2 distributions more clearly.

We now reworked the Figure and hope the results are more clearly presented.

Super-resolution STED microscopy identifies stars and asters in living cells

In this part, the authors use STED microscopy to exclude a role of the fixation procedure in the patterns formation.

- Figures would be improved if the same scale bar was used in the 3 conditions.
- Similarly, the same ROI / zoom and scale bar should be used in the different conditions to highlight the differences in the actin patterns observed.
- Specify the time in Figure 2A

In the previous version of the manuscript, we adjusted the image scaling throughout different Figures to emphasize certain details. We now changed this strategy and present all panels at the same scaling, as requested by the referee.

Dynamic patterning is nucleated and maintained by the Arp2/3 complex

The authors use an additional technique, TIRF-SIM, to observe the dynamics of the patterning formation. They use HeLa cells expressing Lifeact and p16, a sub-domain of the Arp2/3 complex, and they image with a time-lapse of 10s.

1. It is hard to see the dynamics of the Arp2/3 on the filaments in Figure 3. Arrows in images could help and a zoom on some zones of the images.

We now made the scaling consistent for all panels and Figures (see previous comment). Initially, we provided the TP (temporal projections) to emphasize the activity of the Arp2/3 complex but they appeared to be confusing. We therefore removed the TPs from the Figures. We further provide movies supporting the activity of the Arp2/3 complexes.

2. Authors should explain how they define individual Arp2/3 complexes? Co-staining should be presented to show that the probe correctly identifies complex complexes.

We apologize to the reviewer that we did not sufficiently present data for the correct localizations of p16 to the Arp2/3 complexes and for our claims that p16 sub-units were reporting on Arp2/3 complexes. We and others have been using our p16 construct in multiple publications since the time of Professor Charras as a Postdoc at the Mitchison laboratory (Harvard Medical School, USA) whose members have been initially developing and characterizing this construct since the 90's and demonstrated correct localizations at multiple incidences (Welch et al, JCB 138(2): 375-384. 1997; Gournier et al, Mol Cell 2001; Bovellan et al, Current Biology, 2014). We now clarify the origin of the construct and the points raised by the referee in the manuscript.

3. It is not clear from the data that CK666 stops the actin nucleation as stated. It is not enough to state that other studies report a 30% decrease in actin nucleation.

We agree with the reviewer. We now softened this statement in the manuscript.

4. Figure 3B: what is time 0? Is this the time following drug addition?
5. What happens between 30 s (Figure 3B) and 30 min (Figure 3C)? The authors should do a longer time-lapse increasing the total time of the experiment using low laser power, lower exposure time and infrequent imaging.

The treatment time-point refers to the total time of incubation prior imaging of the actin structures. The eTIRF-SIM system unfortunately does not yet technically allow following actin patterns and p16 sub-units over multiple minutes. Currently, it is already challenging to keep the actin structures of interest within the optical focal plane for only a few minutes. We now added a line to the manuscript to clarify these points. From the 30min treatment experiments however, we can expect that the Arp2/3 complexes predominantly remain at the actin structures because we are imaging 30min post treatments and no overall detachment from Arp2/3 was detectable.

6. Why does the Arp2/3 complex distribution shift after addition of the CK666 inhibitor? The inhibitor acts on the Arp2/3 complex by maintaining the complex in the inactive state. How do they explain the accumulation of the Arp2/3 complex on the filaments in the actin asters under these conditions? If the organization of the Arp2/3 complex in this structures is very dynamic, as the authors state in the manuscript, we should observe a loss of Arp2/3 complex from filaments not freezing of movement.

We agree with the reviewer's comment. The majority of the Arp2/3 complexes were located at the centers of the stars and asters but also acted within the surroundings of the patterns. It is not clear why the Arp2/3 complexes accumulate at the peripheral F-actin further away from the pattern cores. Potentially, the Arp2/3 complexes stay attached to the pointed ends of newly formed F-actin, which are located further away from the cores to leave space for de novo forming F-actin during star and aster nucleation. We now speculate about these observations in the discussion.

7. In Figure 3D it is hard to see the dynamics of the Arp2/3 complex. The authors could zoom on the region of interest to make this clearer.

We now made the scaling consistent for all panels and Figures. We also provide movies supporting the activity of the Arp2/3 complexes.

8. Figure 3E: the tracking could be improved. Track 1: the Arp2/3 complex isn't seen across the whole track. Why? How do they distinguish the single molecules? How precisely (over what time period and at what scale in xyz) are aster trajectories determined?

We apologize to the reviewer for this confusion. The representative image in the old Figure 3E (new Figure 5E) showed one time snapshot ($t=0$) of many frames whereas the trajectories of selected Arp2/3 complexes were presented for multiple time frames. We now clarify this in the caption of Figure 5E.

9. TP images are very confusing. The authors could replace these images with representative single trajectories of individual Arp2/3 complexes.

We now removed the majority of the temporal projections. We only kept temporal projects in the Figure 2 to demonstrate the high degree of activity of the peripheral F-actin of vortices which would otherwise be hard to show.

Pattern formation is independent from myosin

The authors investigate the role of the myosinII in the pattern formation. The figures should have the same scale bar, and the images should be better. The authors could add some trajectories to see the effect of the inhibitor and, perhaps, a longer time-lapse to see the dynamics of the myosinII over several minutes?

We now adjusted the scale bars in the Figures. As highlighted before, the eTIRF-SIM system unfortunately does technically not yet allow to follow actin patterns over multiple minutes. Currently, it is still challenging to keep the actin structures in the focal plane for only a couple of minutes. We were therefore technically not able to provide the requested measurements.

Rotating vortices shape cortex assembly

We recommend that the authors start the results by discussing actin organization at shorter times.

We now adjusted the discussion following the referee's suggestion.

General remarks on the observations

How many cells were observed using SEM? This is important as one would expect cells to exhibit heterogeneity like that seen in the fluorescence microscopy experiments?

We now reevaluated and amended the provided statistics such as the cell numbers for all experiments including the SEM experiments in the Materials and Methods sections. We also clarify this point in the manuscript.

Since the spatial resolution is 90 nm in these experiments, perhaps the authors could try to use TIRF instead of TIRF-SIM. With a TIRF microscope one can avoid photo-bleaching and phototoxicity and can decrease exposure time and/or laser power.

Unfortunately, the phototoxicity of TIRF imaging is not strikingly superior to eTIRF-SIM. In addition, TIRF is diffraction-limited (spatial resolution < 200nm laterally) and misses the required spatial resolution to resolve the structural details of asters in particular. While the total aster size is about 1-2um, the aster core can be quite small, which makes it very challenging to identify the patterns within the cortical actin network. Using conventional TIRF, we however imaged very successfully actin stars and larger actin asters but found it difficult to use TIRF for the systematic imaging of the transition between the different patterns and therefore chose not to present the data in the manuscript.

Relation of patterns to podosomes

This part could be better described and discussed.

We now updated this part and added a section to the discussion.

Discussion

The authors describe at some point the patterns observed for microtubules in vitro. In the paper they cite, there are no stars and the transition is from asters to the vortices - not the opposite as the authors describe.

We now clarified this part in the discussion.

Response to Reviewer 2:

The authors report the existence of not so novel actin filament-based structures in HeLa cells. Specifically, asters have been evident in fluorescent phalloidin stains for years. In addition, it is completely unclear what the significance of these structures are to cellular function. There are also several major problems with the manuscript as outlined below.

We thank Reviewer 2 for their careful reading of our manuscript and for agreeing with us that both the significance and functional role of self-organized actin patterns remain elusive, even though some asterisk-shaped actin structures may have been evident in fluorescent images for years. To our best knowledge the self-organization of the actin cortex in form of actin vortices, asters, and stars and their dynamic relationship to each other have not been systematically directly studied and resolved in living cells so far. Most importantly, the significance and functional role of actin self-organization have been argued and demonstrated *in silico* and *in vitro* in many high profile articles throughout different fields in the recent years. Specifically, the impact of cortical actin dynamics including spontaneous pattern formation within the cortical mesh is thought to be essential for the architecture and dynamics of the cellular plasma membrane and cell mechanics. We respectfully refer the reviewer to the articles (Eggeling et al, Nature 2009; Cardamone et al, PNAS 2011; Gowrishankar et al, Cell 2012; Tee et al, Nature Cell Biology 2015), as also outlined in the introduction of our manuscript. We remain confident that our observations are novel and a significant advancement in the understanding of self-organizing actin pattern and genuinely hope that we could convince the reviewer of the importance of our work.

The Reviewer is mainly concerned about the quantification of the percentage of actin patterns present at a given point. She/he also asks for additional evidence for the transitions between the different actin patterns. Although direct observations of the transitions between the different actin patterns is technically very challenging (see response to Reviewer 1), we now added new panels to Fig.1 for the transition from vortices to stars and the new Fig. 7 for transitions between stars and asters to address the referees concerns. Please find below our responses to the detailed comments of the Reviewer 2.

Detailed Comments of Reviewer 2:

(1)The authors need to quantify the percentage of asters and stars at the two time points. Are there absolutely no stars at later time points? Although the authors allude to the rarity of the observed structures (present in only 30% of the cells), quantification would better make the point better that there is transition across the different structures.

We now quantify the number of patterns at a given time and the total surface area occupied by the actin patterns (if present) for all three time-points. We feel that these plots presented in Figure 7D indeed strengthen our arguments and thank the reviewer for this comment.

(2)The biggest pitfall of this paper is that the authors were still not able to show the transition live. Although the authors acknowledge this, why not use TIRF SIM with lower temporal resolution? If they can get 300 total frames, why not do 5 min intervals (gives ~1500 min)?

We addressed the reviewer's concern and now added evidence for the transitions between the observed actin patterns (see also Reviewer 1). Let us however briefly comment on the question raised by Reviewer 2. eTIRF-SIM is probably the only super-resolution technique to-date that allows technically to image the pattern formation at appropriate time- and length-scales as outlined by the reviewer. However, photo-bleaching remains a limitation for these experiments and we were only able to image 300 frames in total. These transitions occur spontaneously over a few seconds only but within a time-frame of multiple hours. Unfortunately, the localization and identification of the structures require continuous imaging and once a transition has been identified, hardly any frames remain to capture the complete transition. Consequently, we blindly imaged the structures and relied on lucky incidences to monitor these transitions. We performed now a significant number of new experiments and could indeed capture some of the transitions but rely on further development of the microscopy technology to image the full sequence of transitions in between the actin patterns at the same high quality as the other images presented in the manuscript. We therefore genuinely hope that the referee acknowledges our additional efforts to respond to her/his concerns. We added a new paragraph to the manuscript to explain the technical limitations of the current microscopy technology in more detail.

(3)The myosin experiment is purely descriptive and there is no quantification. The percentage of asters and stars with myosin absent from the core should be done. The Y drug experiment is also not very informative. What about the tension generation by just binding to actin? Try NMII knockdown and then test whether you still get self-organization of actin.

We thank the reviewer for pointing out the potential 'passive role' of myosin-II in the formation of the actin patterns. We agree with the reviewer that we have not sufficiently clarified in the manuscript that the presence of myosin-II to peripheral actin is potentially essential for the initial formation of the actin patterns. By pharmacologically blocking Rho-kinase with Y27632, our experiments demonstrate that myosin-II is not essential for the nucleation and maintenance of the actin patterns but we cannot exclude that myosin-II may potentially play a role in setting the mechanical environment required for the formation of actin patterns. We now performed more systematically the myosin-II experiments for all actin patterns and adjusted the corresponding sections in the manuscript. We also added additional quantification for the absence of myosin-II foci (Fig. 6E,F) and added a new paragraph to the discussion to clarify the potential role of myosin-II in these processes.

Future studies may want to focus on more biological approaches with systematic protein deletion to investigate how myosin-II and other crosslinking proteins such as fascin, alpha-actinin, and others actin-associating crosslinkers control the intrinsic mechanical stress within the cortical actin to initiate the formation of the self-organized actin patterns. We focused on the nucleation and maintenance of the actin patterns (that appear to be independent of myosin-II). Although we share the views of the referee of such an approach, we fear however that myosin-associated NMII knockdown alone will not be informative about the intracellular stress generation and/or offers no further insights into the role of myosin-II in addition to Y27632 treatments; but NMII knockdown should be part of a series of essential experiments targeting the initiation of self-organized pattern formation. Specifically, constitutive NMII knockdown will severely affect the cell's ability to control its shape, mechanics, and to adhere to the substrate, making potential conclusions on self-organized pattern

organization challenging. All together we feel that these experiments are currently beyond the scope of this study. We therefore respectfully refer the referee to upcoming studies of us and others.

(4)The lipid ordering experiment is also not convincing. The gray levels are not matched and the asters don't really look like asters. Being the only figure that highlights the potential importance of these structures, there should be more quantification, gray level matching.

While the structural organization of asters are easily identifiable in super-resolution images with extended spatial resolution, the confocal images as performed for the lipid ordering experiments are diffraction-limited and miss the required spatial resolution to visualize the nanoscale organization of asters. We now rescaled the images and added a new Panel to Figure 9 (old Figure 7) showing single asters rather than a field of asters at higher magnification to convince the reviewer of the observed changes in the membrane fluidity due to aster formation.

(5)In all of the live cell quantifications in Figure 3, all the Arp2/3 foci are quantified, while the number of actin aster and starts are actually scarce. Maybe another quantification of Arp2/3 mobility within a star or an aster?

To address the referee's suggestion, we added new FRAP experiments to further elucidate the significance of the Arp2/3 complexes locating predominately to the cores of the star and asters (as suggested by Reviewer 1). Surprisingly, we found that there are additional Arp2/3-nucleated F-actin strands situated at the pattern cores (in addition to the peripheral F-actin strands) pointing outwards from the centers of the stars and asters towards the peripheral actin (Figure 5H-J and Figure 5K). These experimental observations are novel and we would therefore like to thank both reviewers for their suggestions that strengthened this part of the manuscript.

(6)The Methods section was hastily written and has more grammatical errors than the rest of the manuscript.

We apologize to the reviewer for these mistakes. We now carefully reworked the Materials and Method section.

REVIEWERS' COMMENTS:

Reviewer #1 (Remarks to the Author):

The authors have worked hard to address the comments of the Reviewers.
The logic of the paper is now better.

The study combines a wide range of techniques (SEM, Live imaging, FRAP, AFM, Image analysis etc) in an attempt to determine the structure and dynamics of cortical actin filaments in HeLa cells, and their influence on cell mechanics and on membrane ordering.

As a result, the paper is very broad and very descriptive.

Many of the findings are interesting but anecdotal as presented (e.g. lipid order/transitions between structures).

There are few conclusions about the function of these networks, only that they are not podosomes. Nevertheless, the results of this analysis will be interesting to compare with in vitro and in silica work.

The conclusion that stands out, is that the Arp2/3 complex rather than Myosin II controls the spatial order and dynamics of cortical actin vortices, asters and stars.

However, as written, the paper remains very hard to follow.

Also, the text includes some very strange sections.

For example, having explained why it is impossible to use FRAP with LifeAct-GFP to study actin filament dynamics, the team go on to use it to study "polarity".

The logic of this is very hard to understand.

Finally, I sincerely apologize to the authors for the delay in this review.

Reviewer #3 (Remarks to the Author):

This manuscript by Fritzsche and colleagues is a beautiful, rigorous, and extensive body of work. It provides the first detailed and comprehensive view of cellular actin cytoskeleton reorganization upon adherence, using a suite of advanced light and electron microscopy approaches. I was fascinated reading the paper by the striking parallels (plus a few important and informative distinctions) between what the authors observe here in living cells and what has been observed in recent in vitro studies on actin self-organization. The cytoskeleton field will benefit greatly from publication of this work, and the authors have done a commendable job of addressing the vast majority of the reviewers' previous suggestions and comments. This thought-provoking study deserves to be in Nature Communications, and is ready for publication.

Reviewers' comments

Reviewer #1

The authors have worked hard to address the comments of the Reviewers. The logic of the paper is now better.

The study combines a wide range of techniques (SEM, Live imaging, FRAP, AFM, Image analysis etc) in an attempt to determine the structure and dynamics of cortical actin filaments in HeLa cells, and their influence on cell mechanics and on membrane ordering. As a result, the paper is very broad and very descriptive. Many of the findings are interesting but anecdotal as presented (e.g. lipid order/transitions between structures).

There are few conclusions about the function of these networks, only that they are not podosomes. Nevertheless, the results of this analysis will be interesting to compare with in vitro and in silica work. The conclusion that stands out, is that the Arp2/3 complex rather than Myosin II controls the spatial order and dynamics of cortical actin vortices, asters and stars.

We thank the reviewer for the positive evaluation of our manuscript. In the revised version we have changed the text to make it less descriptive and anecdotal – the text has also been shortened.

However, as written, the paper remains very hard to follow. Also, the text includes some very strange sections. For example, having explained why it is impossible to use FRAP with LifeAct-GFP to study actin filament dynamics, the team go on to use it to study "polarity". The logic of this is very hard to understand.

We have revised the text to make it easier to follow. Especially we have shortened the main text, moving some sections of the supplements. Also we have rephrased some strange sections such as the section on the FRAP data, which now also includes a few more introductory sentences.

Finally, I sincerely apologize to the authors for the delay in this review.

Apologizes taken.

Reviewer #3

This manuscript by Fritzsche and colleagues is a beautiful, rigorous, and extensive body of work. It provides the first detailed and comprehensive view of cellular actin cytoskeleton reorganization upon adherence, using a suite of advanced light and electron microscopy approaches. I was fascinated reading the paper by the striking parallels (plus a few important and informative distinctions) between what the authors observe here in living cells and what has been observed in recent in vitro studies on actin self-organization. The cytoskeleton field will benefit greatly from publication of this work, and the authors have done a commendable job of addressing the vast majority of the reviewers' previous suggestions and comments. This thought-provoking study deserves to be in Nature Communications, and is ready for publication.

We thank the reviewer for his/her utmost positive evaluation of our manuscript.